# Variance-Guided Score Regularization for Hallucination Mitigation in Diffusion Models

## Abstract

Diffusion models have emerged as the backbone of modern generative AI, powering advances in vision, language, audio and other modalities. Despite their success, they suffer from *hallucinations*, implausible samples that lie outside the support of true data distribution, which degrade reliability and trust. In this work, we provide a density-based perspective on hallucinations and derive an explicit lower bound linking *score smoothness* to nonzero probability mass allocated to gap regions outside the data support leading to a positive hallucination rate. To mitigate hallucinations, we introduce a *Variance-Guided Score Regularization* (VSR) strategy that explicitly controls the score Jacobian, thereby reducing the lower bound on gap mass. Empirical results on synthetic and real-world datasets demonstrate that our approach reduces hallucinations ($\sim 25\%$) while maintaining high fidelity and diversity, providing a principled step toward more reliable diffusion-based image generation. We also propose two benchmark datasets with controlled semantic variation for systematic hallucination evaluation.

## 1 Introduction

Diffusion models Song et al. (2020a); Rombach et al. (2022); Ho et al. (2020b); Nichol & Dhariwal (2021) have become the de facto backbone of multi-modality generation. They have been widely used in image synthesis Rombach et al. (2022); Saharia et al. (2022), audio generation Kushwaha et al. (2025), text synthesis Wu et al. (2023); Li et al. (2022), and biomedical applications Guo et al. (2024); Bhosale et al. (2025). Recent test-to-image systems FLUX Kontext Labs et al. (2025), Nano-Banana Google DeepMind & Google Research (2025), and Stable Diffusion 3.5 Stability AI (2024) have pushed fidelity, controllability, and latency, enabling interactive editing. Adoption is accelerating at scale: within the span of two years, Adobe Firefly reports 22B+ assets generated as of April 2025 Adobe (2025), and enterprise AI usage broadly rose to $78\%$ of organizations in 2024 Stanford HAI (2025).

While diffusion-based text-to-image systems are widely adopted, they raise well-documented concerns around fairness/bias, content safety, privacy, and copyright issues Huang et al. (2025); Hao et al. (2023); Devulapally et al. (2025). In this work we focus on *hallucinations* : implausible samples generated by diffusion models (e.g., human hands with extra or missing fingers) Aithal et al. (2024); Oorloff et al. (2025).

Beyond reducing sample quality, hallucinations undermine trust in the reliability of model generations. However, hallucinations in diffusion models are still largely underexplored. Kim et al. (2024) mitigate structural hallucination in image translation with multiple local diffusion. However, they do not use common text conditioned image generation setup. Aithal et al. (2024) study hallucination as mode interpolation but the work does not propose any hallucination mitigation strategies. Oorloff et al. (2025) proposes to use temperature scaled self attention, but do not propose mitigation in text-conditioned image generation setting. In this work, We formalize a density-based view of hallucinations and introduce a simple, training-time regularizer grounded in a derived lower bound on off-support probability mass.

Our **key contributions** are: (i) We derive an explicit point-wise lower bound on learned model density (using Diffusion Models) outside data support, showing it decays exponentially away from

---

[2]Equal advising.

data support and is governed by the score function's [1] local bound and Lipschitz constant. (ii) We introduce a effective, architecture-agnostic smoothness regularizer that increases the local score curvature (via the Jacobian of score function), which tightens the theoretical lower bound on the gap mass. We give a tractable instantiation guided by learned denoising variance. (iii) We propose two challenge datasets with extreme (very large number ($> 10^{44}$) of) semantic classes (*ChessImages*, *Cards*) to probe hallucination in controlled settings. On three established benchmarks, our method reduces hallucinations by $\sim 15\%$, and on the proposed benchmarks by $\sim 25\%$ compared to baselines.

## 2 RELATED WORK

**Hallucinations.** Kim et al. (2024) is one of the first hallucination mitigation work in diffusion-based proposing that proposes "Local Diffusion" pipeline that estimates OOD regions, performs branched denoising inside/outside the mask, and fuses the results. However, they require expert mask annotation for medical datasets to construct OOD region. VSR does not depend on any additional annotation. Aithal et al. (2024) introduce hallucinations as explained by mode interpolation: interpolating between disjoint modes due to smooth learned score approximations. But this work does not propose any mitigation technique or theoretical bound. Oorloff et al.Oorloff et al. (2025) change temperature self-attention mechanism. They use temperature scaling in attention softmax to adaptively suppress early-stage noise contributing to hallucinations. Lu et al. (2025) frame text hallucination as a local generation bias, introduce the Local Dependency Ratio (LDR) to measure it, and argue that stronger global dependencies help. However, their analysis is only focused on images containing text.Wewer et al. (2025) reduce hallucinations on structured reasoning tasks by sequentializing diffusion generation with Spatial Reasoning Models (SRMs), learning a variable order. But SRMs only target on datasets with spatial reasoning and hence limited in their applicability to widely used text-to-image generation. They also propose MNIST Sudoku dataset, but we highlight that ChessImages dataset proposed in this work has higher semantic classes ($\sim 10^{44}$ vs $\sim 10^{22}$).

## 3 HALLUCINATIONS IN DIFFUSION MODELS

We now formalize the notion of hallucinations in the context of Diffusion Models (DMs). Combining notations from Aithal et al. (2024) and Pham et al. (2025), we categorize generated samples $\tilde{x} \sim \mathcal{P}_\theta$ into: (i) **Hallucinated** and (ii) **Non-Hallucinated**. We further sub-categorize non-hallucinated samples into (i) **Memorized** and (ii) **Generalized** Samples.

---

**Definition 1: Hallucinated Samples**

Let $\mathcal{P}_{\text{data}}$ denote the true data distribution over $\mathcal{X} \subseteq \mathbb{R}^d$, and let $\mathcal{P}_\theta$ be the learned generative distribution by a diffusion model with parameters $\theta$. We use $p_{\text{data}}(x)$ and $p_\theta(x)$ to denote their corresponding density functions. A generated sample $\tilde{x} \sim \mathcal{P}_\theta$ is said to be **hallucinated** if it lies outside the support of true data distribution. Formally, for a small tolerance $\epsilon > 0$, the $\epsilon$-hallucination set is defined as

$$\mathcal{H}_\epsilon(\mathcal{P}_{\text{data}}) := \{ \tilde{x} : p_{\text{data}}(\tilde{x}) \leq \epsilon \}.$$

A generated sample $\tilde{x} \sim \mathcal{P}_\theta$ is hallucinated if $\tilde{x} \in \mathcal{H}_\epsilon(\mathcal{P}_{\text{data}})$.

---

**Definition 2: Hallucination Rate**

For a generative model with distribution $\mathcal{P}_\theta$ (density $p_\theta(x)$) and true data distribution $\mathcal{P}_{\text{data}}$, the *hallucination probability* is defined as

$$\mathbb{P}_\theta^{\text{hall}}(\epsilon, \mathcal{P}_{\text{data}}) := \Pr_{\tilde{x} \sim \mathcal{P}_\theta} [\tilde{x} \in \mathcal{H}_\epsilon(\mathcal{P}_{\text{data}})] = \int_{\mathcal{H}_\epsilon(\mathcal{P}_{\text{data}})} p_\theta(x) \, dx,$$

where $dx$ denotes integration w.r.t. the Lebesgue measure Folland (1999); Bishop (2006).

---

[1]score function is defined as the gradient of the log probability density

**Non-Hallucinated Samples.** A generated sample $\tilde{x} \sim \mathcal{P}_\theta$ is *non-hallucinated* if it does not belong to the hallucination set, i.e., $\tilde{x} \notin \mathcal{H}_\epsilon(\mathcal{P}_{\text{data}})$. Within this class, following Pham et al. (2025), we further distinguish between *memorized* and *generalized* samples.

---

**Definition 3: Memorized vs. Generalized Samples**

Let $d(\cdot, \cdot)$ be a similarity metric on $\mathcal{X} \subset \mathbb{R}^d$, $\delta > 0$ a small tolerance, and $\mathcal{X}_{\text{train}} = \{x^{(1)}, \ldots, x^{(N)}\}$ the training set. For a generated sample $\tilde{x} \sim \mathcal{P}_\theta$:

$$\mathcal{M} := \{ \tilde{x} \mid \exists x^{(i)} \in \mathcal{X}_{\text{train}}, \; d(\tilde{x}, x^{(i)}) \leq \delta \}, \quad \mathcal{G} := \{ \tilde{x} \mid \tilde{x} \notin \mathcal{M}, \; \tilde{x} \notin \mathcal{H}_\epsilon(\mathcal{P}_{\text{data}}) \}.$$

Here $\mathcal{M}$ are memorized (near-duplicate) samples, while $\mathcal{G}$ are generalized samples: novel yet valid points in the high-probability region of $\mathcal{P}_{\text{data}}$.

---

In this work, we propose a method to mitigate the number of hallucinated samples [2]. We will now see the effect of score smoothness on the hallucination rate.

## 4 RELATION BETWEEN SCORE SMOOTHNESS AND HALLUCINATION PROBABILITY

Although score smoothness in diffusion models has been widely studied Ho et al. (2020a); Song et al. (2020b); Aithal et al. (2024), our contribution is to show that this smoothness induces a nonzero hallucination rate. The Gaussian forward process produces $C^\infty$, positive densities Arendt (2005); Evans (2010), and discrete-time DDPM transitions remain nondegenerate Gaussians Ho et al. (2020a). As a result, diffusion marginals assign positive probability to any open region of $\mathcal{X}$, including "gaps" outside the true data support, directly linking smooth score dynamics to hallucinations (Sec. 3).

**Preliminaries.** Let $\mathcal{X} \subseteq \mathbb{R}^d$ denote the data space, and let $x_0 \sim p_{\text{data}}$ be a clean data sample. The *score function* Song et al. (2020b) is $s(x) = \nabla_x \log p(x)$. In the forward diffusion process, data is corrupted by Gaussian noise: $q(x_t \mid x_0) = \mathcal{N}(\sqrt{\bar{\alpha}_t}\, x_0, \; (1 - \bar{\alpha}_t)I)$ where $t$ indexes the noise level. The ground-truth score at time $t$ is

$$s_{\text{GT}}(x_t, t) = \nabla_{x_t} \log q(x_t \mid x_0) = -\frac{x_t - \sqrt{\bar{\alpha}_t}\, x_0}{1 - \bar{\alpha}_t} = -\frac{\epsilon}{\sqrt{1 - \bar{\alpha}_t}},$$

showing that the score corresponds to the added noise $\epsilon$, rescaled by, $-(1 - \bar{\alpha}_t)^{-1/2}$.

In practice, the model learns an approximation $s_\theta(x_t, t)$, trained to match $s_{\text{GT}}(x_t, t)$ across timesteps Song et al. (2020b). We measure the approximation error per dimension and its root-mean-squared aggregate as

$$\Delta s_d(x_t, t) = s_{\theta, d}(x_t, t) - s_{\text{GT}, d}(x_t, t), \qquad \Delta s = \sqrt{\frac{1}{TD} \sum_{t=1}^{T} \sum_{d=1}^{D} \big(\Delta s_d(x_t, t)\big)^2}, \qquad (1)$$

which reports the average score error across time and dimensions. This error quantifies how well the learned score preserves the ground-truth score.

**Setup.** Let $p_{\text{data}}$ denote the true data distribution on $\mathcal{X}$. Assume $p_{\text{data}}$ decomposes into $K$ disjoint components with compact supports that are $\delta$-separated:

$$p_{\text{data}} = \sum_{k=1}^{K} w_k\, p_k, \; \text{supp}(p_k) \cap \text{supp}(p_\ell) = \emptyset \; (k \neq \ell), \; \text{dist}\big(\text{supp}(p_k), \text{supp}(p_\ell)\big) \geq \delta > 0,$$

where $\text{supp}(p_k)$ denotes the *support* of $p_k$, i.e., the smallest closed set of full measure under $p_k$. Equivalently, when $p_k$ admits a density, $\text{supp}(p_k) = \{ x \in \mathbb{R}^d \mid p_k(x) > 0 \}$.

We define the *gap region* as $G := \mathcal{X} \setminus \bigcup_{k=1}^{K} \text{supp}(p_k), \; p_{\text{data}}(x) = 0$ for all $x \in G$.

Let $p_\theta$ denote the density of a trained diffusion model. Because the Gaussian corruption enforces smooth scores and positive densities Song et al. (2020b); Ho et al. (2020a); Yang et al. (2024), the

---

[2] This work does not aim at increasing the count of generalized samples. We aim to observe the effect of score smoothness on the hallucination rate

model inevitably assigns nonzero probability mass to $G$. This motivates our method for reducing hallucinations by explicitly controlling the smoothness-to-hallucination relationship.

**Lemma 1** (Positivity of Model Density Implies Hallucinations). *In diffusion models, each reverse transition is a nondegenerate Gaussian with $\Sigma_\theta(x_t, t) \succ 0$ Ho et al. (2020a); Song et al. (2020b), hence the terminal density $p_\theta(x)$ is continuous and positive for all $x \in \mathbb{R}^d$.*

*Let $G \subset \mathbb{R}^d$ be any open gap region disjoint from the true data support with $\lambda^d(G) > 0$. Then*

$$p_\theta(G) = \int_G p_\theta(x)\, dx > 0, \qquad \implies \qquad \mathbb{P}_\theta^{\text{hall}}(\epsilon) \geq p_\theta(G) > 0$$

*for any sufficiently small $\epsilon > 0$. For a positive d-dimensional lebesque measure $\lambda^d$*

Thus, hallucinations are existent in diffusion models with nondegenerate Gaussian reverse kernels. In the next section, we provide a quantitative lower bound (Theorem 2) linking *score smoothness* to the probability mass that leaks into $G$.

**Regularity properties and assumption near the gap boundary.** To derive a quantitative lower bound on the hallucination probability, we utilize certain regularity conditions of the learned diffusion density $p_\theta$ near the boundary of the gap region $G$. Let $\partial G$ denote the boundary of $G$. We rely on the following properties (P) and assumptions (A):

(P1) **Positivity and smoothness.** The model density $p_\theta$ is continuous, $C^1$, and positive on $\mathbb{R}^d$, as guaranteed by the Gaussian reverse kernels in DDPMs and score-based models (Lemma 1).

(P2) **Local Lipschitz continuity of the score.** The score $s_\theta(x) := \nabla \log p_\theta(x)$ is $L$-Lipschitz in a tubular neighborhood

$$U := \{x \in \mathbb{R}^d : \text{dist}(x, \partial G) \leq r\}, \quad r > 0,$$

i.e., $\|s_\theta(x) - s_\theta(y)\| \leq L\|x - y\|$ for all $x, y \in U$. This follows from the Gaussian smoothing effect in the forward process, which induces smooth and Lipschitz-continuous scores for all $t > 0$ Song et al. (2021); Yang et al. (2024).

(P3) **Boundedness of the score.** On compact neighborhoods such as $U$, the score remains bounded,

$$\sup_{x \in U} \|s_\theta(x)\| \leq S < \infty,$$

This is a direct consequence of the Gaussian perturbation Song et al. (2021), which prevents blow-up of the score for $t > 0$.

(A4) **Boundary density lower bound.** Assume $p_\theta$ is continuous and positive everywhere, and $\partial G$ is compact. With Extreme Value Theorem that $p_\theta$ attains a minimum on $\partial G$. We have a uniform lower bound on the model density at the gap boundary Rudin (1976b); Royden & Fitzpatrick (2010): $\inf_{z \in \partial G} p_\theta(z) = C_b > 0$.

Together, (P1)–(P3) and (A4) capture the fact that diffusion models produce smooth, positive, and well-behaved densities around the boundary of the data support, making them amenable to quantitative analysis of hallucination probability.

**Theorem 2** (Smooth score leads to positive mass in gaps with an explicit bound). *Let $x \in G$ with $\delta_x := dist(x, \partial G) \leq r$ and let $y \in \partial G$ be a nearest point to $x$. Under (P1)–(P3) and (A4),*

$$\log p_\theta(x) \geq \log p_\theta(y) - S\,\delta_x - \tfrac{L}{2}\,\delta_x^2, \qquad \text{hence} \qquad p_\theta(x) \geq C_b \exp\left(-S\,\delta_x - \tfrac{L}{2}\,\delta_x^2\right) > 0.$$

*In particular, $p_\theta(x) > 0$ for all $x$ with $\delta_x \leq r$, and the right-hand side depends explicitly on the score's Lipschitz constant $L$ and its local bound $S$.*

*Proof.* Let $f := \log p_\theta$. Since $s_\theta = \nabla f$ is $L$-Lipschitz on $U$, the standard descent lemma gives

$$f(x) \geq f(y) + \langle \nabla f(y), x - y \rangle - \tfrac{L}{2}\,\|x - y\|^2.$$

By Cauchy–Schwarz Rudin (1976a) and (P3), $\left|\langle \nabla f(y), x - y \rangle\right| \leq \|\nabla f(y)\|\,\|x - y\| \leq S\,\delta_x$, and with $\|x - y\| = \delta_x$ and (A4) we obtain

$$\log p_\theta(x) \geq \log p_\theta(y) - S\delta_x - \tfrac{L}{2}\,\delta_x^2 \implies p_\theta(x) \geq p_\theta(y)\, e^{(-S\delta_x - \tfrac{L}{2}\delta_x^2)} \geq C_b\, e^{(-S\delta_x - \tfrac{L}{2}\delta_x^2)} > 0.$$

$\square$

**Interpretation.** Theorem 2 shows that if the score is Lipschitz Song et al. (2020b); Song & Ermon (2019) and locally bounded Song et al. (2020b), then the log-density at any $x$ inside a gap is controlled by its boundary value up to a linear (scaled by $S$) and a quadratic (scaled by $L$) term in $\delta_x$. Consequently, $p_\theta(x)$ admits an explicit positive lower bound depending on $L$, $S$, and $C_b$, ensuring strictly positive mass within distance $r$ of $\partial G$.

By aggregating the pointwise lower bound from Theorem 2 over the neighborhood $G_r := \{x \in G : \mathrm{dist}(x, \partial G) \leq r\}$, we obtain a positive lower bound on the total mass assigned by $p_\theta$ to this region. Relating this region to the empirical threshold that defines hallucinations (recall $\mathcal{H}_\epsilon(\mathcal{P}_{\mathrm{data}})$ from section 3), we obtain a quantitative lower bound on the hallucination rate.

**Corollary 1** (Positive Lower Bound leads to non-zero Hallucination rate). *Let $G \subset \mathcal{X}$ be the support of $\mathcal{P}_{\mathrm{data}}$, and define $G_r := \{x \in G : \mathrm{dist}(x, \partial G) \leq r\}$. By Theorem 2, for all $x \in G_r$*

$$p_\theta(x) \geq C_b \, e^{-Sr - \frac{L}{2}r^2}, \quad \text{hence} \quad \int_{G_r} p_\theta(x)\, dx \geq \lambda^d(G_r)\, C_b\, e^{-Sr - \frac{L}{2}r^2} > 0$$

*where $\lambda^d$ is the $d$-dimensional Lebesgue measure Rudin (1976a).*

*Proof.* By Definition 3, $\mathcal{H}_\epsilon(\mathcal{P}_{\mathrm{data}}) := \{x : p_{\mathrm{data}}(x) \leq \epsilon\}$. Let $\varepsilon_r := \sup_{x \in G_r} p_{\mathrm{data}}(x)$. Then for any $\epsilon \geq \varepsilon_r$, $G_r \subseteq \mathcal{H}_\epsilon(\mathcal{P}_{\mathrm{data}})$, so

$$\mathbb{P}_\theta^{\mathrm{hall}}(\epsilon, \mathcal{P}_{\mathrm{data}}) = \int_{\mathcal{H}_\epsilon(\mathcal{P}_{\mathrm{data}})} p_\theta(x)\, dx \geq \int_{G_r} p_\theta(x)\, dx \geq \lambda^d(G_r)\, C_b\, e^{-Sr - \frac{L}{2}r^2} > 0.$$

$\square$

**Takeaway.** Lemma 1 shows that Gaussian reverse dynamics already force $p_\theta$ to place nonzero mass outside the data support Ho et al. (2020a); Song et al. (2020b). Theorem 2 strengthens this by giving an *explicit* pointwise lower bound inside the gap in terms of the score's local boundedness and Lipschitz constant (properties induced by Gaussian smoothing and the reverse-time score formulation Song et al. (2021; 2020b); Yang et al. (2024)). Corollary 1 then lifts the pointwise bound to a *quantitative* lower bound on total mass in the tubular gap region $G_r$, and hence on the hallucination rate defined in Sec. 3.

## 5 REDUCING HALLUCINATION VIA SMOOTHNESS REGULARIZER

Having shown in Theorem 2 that *smooth* scores impose a positive gap floor and, via Corollary 1, a positive hallucination probability (Sec. 3), we now introduce a score smoothness regularizer that reduces the lower bound on gap mass at every timestep.

**Smoothness regularizer.** We augment the denoising objective with a time-dependent smoothness penalty

$$\mathcal{L}_{\mathrm{smooth}} = \mathbb{E}_{t, x_t}\Big[ \phi\left(\|J_\theta(x_t, t)\|^2\right) \Big], \qquad \phi(u) = \frac{1}{u + \eta}, \; \eta > 0, \tag{2}$$

and train with $\mathcal{L}_{\mathrm{tot}} = \mathcal{L}_{\mathrm{DSM}} + \rho\, \mathcal{L}_{\mathrm{smooth}}$ where $\rho > 0$ is the regularization strength, $\mathcal{L}_{\mathrm{DSM}}$ is a standard denoising matching loss Ho et al. (2020a). Minimizing equation 2 encourages larger $\|J_\theta\|$ on the support of $(t, x_t)$. On tube $U$ (P2) since the (vector-valued) mean value theorem on $U$ implies $L_\theta(t) = \sup_{z \in U} \|J_\theta(z, t)\| \geq \|J_\theta(z, t)\|$ $(\forall z \in U)$ for any consistent choice of norms Clarke & Ledyaev (1994). Therefore $\mathcal{L}_{\mathrm{smooth}}$ tightens the lower bound.

**Implementation of $\mathcal{L}_{\mathrm{smooth}}$ in practice:** Recent works Meng et al. (2021) have shown that higher-order derivatives of $\log p(x)$, in particular the Jacobian of the score function, encode crucial information about the conditional variance of the "clean" data given a noisy observation. Meng et al. (2021) derive methods to estimate second-order scores (i.e. $H_f(x) = \nabla^2 \log p(x)$) and show that these are directly related via generalized Tweedie's formula to posterior covariance in Gaussian denoising settings. Let the forward process be $q(\tilde{x} \mid x) = \mathcal{N}(\alpha x, \sigma^2 I)$. The posterior $q(x \mid \tilde{x})$ is also Gaussian with covariance $\Sigma(\tilde{x}) = \frac{\sigma^4 \nabla_{\tilde{x}}^2 \log q(\tilde{x}) + \sigma^2 I}{\alpha^2}$. Using Bayes' rule, the score and Hessian satisfy: $\nabla_x \log p(x) = -\Sigma^{-1}(x - \mu)$, $\quad \nabla s_\theta(x) = \nabla_x^2 \log p(x) = -\Sigma^{-1}$. We use denoising covariance to implement $\mathcal{L}_{smooth}$ in our work. Calculating the high-dimensional covariance of the denoising process is intractable; therefore, we depend on the methods that learn the diagonal covariance of the denoising data distribution. In particular, we employ the I-DDPM Nichol & Dhariwal (2021) implementation to learn covariance.

| Dataset | Detection Type | Time (100 images) | RGB | Size | Semantic Classes |
|---|---|---|---|---|---|
| HandsAfifi (2019) | Human Annotation | $\sim$12 min | ✓ | 11,079 | 4 |
| ShapesAlaa et al. (2022) | Training-Free | $\sim$2.5 s | ✗ | 22,000 | 6 |
| MNISTLecun et al. (1998) | Classifier | $\sim$4 s | ✗ | 60,000 | 10 |
| **Cards (Proposed)** | Training-Free | $\sim$2.5 s | ✓ | 94,000 | $21 \times 10^5$ |
| **ChessImages (Proposed)** | Training-Free | $\sim$2.5 s | ✓ | 190,000 | $10^{44}$ |

Table 1: Characteristics of datasets used. A *semantic class* denotes a valid, interpretable configuration. The proposed *Cards* and *ChessImages* datasets feature vast semantic spaces and allow rapid training-free hallucination detection ($\sim$2.5s per 100 images), making them effective benchmarks for systematic hallucination studies.

# 6 EXPERIMENTS

We evaluate the proposed smoothness regularizer $\mathcal{L}_{\text{smooth}}$ on five high-dimensional datasets (synthetic and real-world). Additionally, Effectiveness of our method is demonstrated both through reduction of score difference ($\Delta S$) on synthetic 1D/2D benchmarks with known ground-truth scores, and through analysis on larger datasets. We first describe the hallucination detection module ($\mathcal{D}$) in section 6.1, then detail the datasets in section 6.2, followed by the experimental setup and metrics (section 6.3), and results (section 7).

## 6.1 HALLUCINATION DETECTION MODULE

To operationalize Definition 3, we introduce a *hallucination detection module* $\mathcal{D} : \mathcal{X} \to \{0, 1\}$ that classifies each generated sample $\tilde{x} \sim \mathcal{P}_\theta$ as hallucinated ($\mathcal{D}(\tilde{x}) = 1$) or valid ($\mathcal{D}(\tilde{x}) = 0$). Instantiations include: (i) human annotation, (ii) classifier-based thresholds, and (iii) training-free rule- or distribution-matching checks. We calibrate $\mathcal{D}$ such that for all $x \sim \mathcal{P}_{\text{data}}$ from training, test sets, $\Pr[\mathcal{D}(x) = 1 \mid x \sim \mathcal{P}_{\text{data}}] = 0$, ensuring genuine samples are not mislabeled. This ensures that detections primarily reflect spurious generations from $\mathcal{P}_\theta$ rather than detector bias.

## 6.2 DATASETS

We evaluate on both standard benchmarks and two new datasets designed for hallucination analysis. A key attribute is the number of **semantic classes**: the set of structurally valid, interpretable configurations each sample belongs to. Constrained datasets (e.g., MNIST with 10 digits) offer limited class spaces, whereas combinatorial datasets (e.g., the proposed *ChessImages* dataset, $\sim 10^{44}$ board states) present extreme diversity, making hallucinations easier to surface. Our proposed *Cards* and *ChessImages* datasets combine (i) extremely large semantic spaces and (ii) efficient, training-free validators, enabling systematic hallucination analysis at scale.

### 6.2.1 DATASETS WITH SIMPLE SEMANTIC CLASSES

*1D and 2D Gaussian Distributions:* For 1D, we sample from a three-mode Gaussian with means $1.0, 1.5, 2.0$ and $\sigma = 0.035$. For 2D, we use a $5 \times 5$ grid of 2D Gaussians with $\sigma = 0.02$. The data support is $\pm 6\sigma$ around each mean in both cases Aithal et al. (2024). Therefore, any generated sample outside this range is considered hallucinated. We train on $5 \times 10^5$ data points and draw $10^6$ samples at inference. We provide additional results on 1D and 2D datasets in the Appendix B.

*MNIST* comprises of $28 \times 28$ grayscale digit images (0–9) Lecun et al. (1998). We label a generated image as a hallucination if it does not resemble any digit. A CNN trained on MNIST (99.5%+ test accuracy) flags outputs with classifier confidence below $0.98$. Number of semantic classes are $10$, one for each digit present.

*Shapes* contains $64 \times 64$ images split into three vertical regions, each assigned a square, a pentagon, or a triangle Aithal et al. (2024). Valid images have at most one shape in each region leading to 6 possible semantic classes. Hallucinations are shapes in the wrong region, duplicates, or blank images. A template-matching pipeline achieves 100% region-and-shape accuracy on training data, enabling automatic hallucination detection.

| Method | H% (1D) ↓ | H% (2D) ↓ |
|---|---|---|
| DDPM[†] | $0.0052173 \pm 0.00192$ | $1.1844 \pm 0.0108$ |
| + VSR $\mathcal{L}_{\text{smooth}}$ | **0.0027027 ± 0.000863** | **1.0831 ± 0.00823** |

Table 2: Effect of $\mathcal{L}_{\text{smooth}}$ on synthetic Gaussian datasets: H% decreases in both 1D, 2D with $10^6$ gen. samples. DDPM[†] Nichol & Dhariwal (2021)

| Method | ΔScore RMSE ↓ | ΔScore $L_2$ Norm ↓ | H% ↓ |
|---|---|---|---|
| DDPM[†] | $21.92 \pm 0.57$ | $7448.96 \pm 146.54$ | $11.00 \pm 2.37$ |
| + VSR $\mathcal{L}_{\text{smooth}}$ | **15.49 ± 0.29** | **5835.22 ± 64.40** | **5.01 ± 1.98** |

Table 3: Hands dataset results: VSR reduces score error ($\Delta S$) and hallucination rate $H\%$, confirming improved sample validity. DDPM[†] Nichol & Dhariwal (2021)

*Hands* includes $128 \times 128$ images of human hands with exactly 5 fingers Afifi (2019). Hallucinations in the hands contain images with missing, extra, or malformed fingers. Three human annotators manually identified hallucinations. Total number of semantic classes in Hands is 2 orientation $\times 2$ (palm up vs palm down) $= 4$.

### 6.2.2 Datasets with Extreme Semantic Class Spaces

*Cards* is a synthetic dataset of $128 \times 128$ images, each arranged in a $2 \times 2$ grid of numbered playing cards (Ace–10). Symbols are rendered in canonical colors (red for hearts/diamonds, black for spades/clubs) using Wikipedia templates.[3] The dataset spans $\sim 21 \times 10^5$ semantic classes. A generation is labeled hallucinated if (i) symbol count mismatches the card value, (ii) symbol color is incorrect, (iii) symbols are missing/invalid, or (iv) conflicting symbols appear. Detection is fully automated via a lightweight template-matching pipeline that verifies symbol type, count, and color, achieving $100\%$ accuracy on the training set and scaling efficiently to large generated samples.

*ChessImages* contains $256 \times 256$ images of legal chessboard configurations rendered from Forsyth–Edwards Notation (FEN) strings sampled from the *VALUED Dataset* Saha et al. (2023). Piece visuals use standardized Wikipedia templates.[4] The dataset has $\sim 190{,}000$ samples, a practical subset of the $(4.822 \pm 0.028) \times 10^{44}$ valid board states Tromp & Österlund (2022). Hallucination detection involves reconstructing FEN from rendered images via template matching ($100\%$ accurate on the dataset), followed by rule validation using `python-chess`. Example samples are shown in Appendix fig. 10.

### 6.3 Experimental Setup and Evaluation Metrics

We evaluate hallucination reduction in pixel-space diffusion (DDPM Nichol & Dhariwal (2021)) and latent-space diffusion (LDM Rombach et al. (2022)), adding $\mathcal{L}_{\text{smooth}}$ as a regularizer. For LDM we test three variants: unconditional (LDM-UC), conditional (LDM-C), and conditional with prompt tuning (LDM-C PT Mahajan et al. (2024)). Prompt tuning details are in the Appendix I. We also compare against AAM Oorloff et al. (2025), the only prior hallucination-mitigation approach. All reported results are averaged over six random seeds.

We report fidelity and hallucination rate across datasets. Fidelity is measured by FID (Inception features) and C-FID (CLIP features). Hallucination rate is given as percentage of detected hallucinations. To assess generalization vs. memorization, we use FLD Jiralerspong et al. (2023), which jointly measures novelty, diversity, and fidelity (computed only on non-hallucinated samples). On *ChessImages*, FLD enables fine-grained generalization analysis. We also show that our method can nearly eliminate hallucinations on the *Cards* dataset.

## 7 Results

**VSR reduces score error** $\Delta S$**:** On synthetic 1D and 2D Gaussian data with known score functions, incorporating $\mathcal{L}_{\text{smooth}}$ reduces hallucinations by $\sim 50\%$ (1D) and $\sim 10\%$ (2D), confirming that controlling score smoothness mitigates leakage in low-density regions. On real-world data, Table 3 shows similar trends: both $\Delta S$ and hallucination rate ($H\%$) decrease on the Hands dataset.

---

[3] https://en.wikipedia.org/wiki/Playing_card
[4] https://en.wikipedia.org/wiki/Template:Chess_diagram

| Method | Hands-11K | | | | MNIST | | | |
|---|---|---|---|---|---|---|---|---|
| | C-FID ↓ | FID ↓ | FLD ↓ | H% ↓ | C-FID ↓ | FID ↓ | FLD ↓ | H% ↓ |
| DDPM[†] | 12.00 | 126.25 | 35.99 | 23.33 | 16.23 | 112.16 | 28.14 | 4.50 |
| + VSR $\mathcal{L}_{\text{smooth}}$ | **10.13** | **108.12** | **22.20** | **5.15** | **8.47** | **43.75** | **8.18** | **3.17** |
| LDM-UC[‡] | 8.89 | 45.78 | 24.87 | 19.66 | 11.82 | 76.98 | 25.29 | 1.83 |
| + VSR $\mathcal{L}_{\text{smooth}}$ | **7.75** | **43.98** | **22.21** | **16.54** | **3.91** | **31.38** | **6.28** | **0.33** |
| LDM-Text Condition[‡] | 10.02 | 83.96 | 21.34 | 29.50 | **8.89** | 230.13 | 23.59 | 23.00 |
| + VSR $\mathcal{L}_{\text{smooth}}$ | **5.58** | **44.95** | **20.07** | **21.15** | 9.36 | **228.21** | **8.74** | **12.48** |
| LDM-PT[‡] | 10.17 | 44.15 | 24.20 | 24.83 | 8.44 | 64.27 | 23.58 | 19.83 |
| AAM[††] | – | 102.30 | – | 9.20 | – | 15.10 | – | 5.70 |
| **Method** | **Cards** | | | | **Shapes** | | | |
| | C-FID ↓ | FID ↓ | FLD ↓ | H% ↓ | C-FID ↓ | FID ↓ | FLD ↓ | H% ↓ |
| DDPM[†] | 9.10 | 112.33 | 33.29 | 22.41 | 26.07 | 123.34 | 21.84 | 29.50 |
| + VSR $\mathcal{L}_{\text{smooth}}$ | **2.20** | **64.35** | **21.40** | **2.33** | **18.98** | **98.61** | **17.29** | **3.00** |
| LDM-UC[‡] | 7.28 | 87.53 | 42.54 | 17.60 | 2.04 | 24.42 | 9.74 | 7.17 |
| + VSR $\mathcal{L}_{\text{smooth}}$ | **3.78** | **32.54** | **19.35** | **7.60** | **1.56** | **19.84** | **7.04** | **4.67** |

Table 4: VSR $\mathcal{L}_{\text{smooth}}$ reduces hallucinations relative to baselines across Hands-11K, MNIST, Cards, and Shapes. Metrics: C-FID = CLIP FID, FID = Inception FID, FLD = Feature Likelihood Divergence, H% = hallucination rate. "–" = unavailable. [†] Nichol & Dhariwal (2021). [‡] Rombach et al. (2022). [††] Oorloff et al. (2025) (values from ArXiV, public code unavailable).

**VSR reduces hallucinations on image datasets:** As shown in table section 7, VSR consistently lowers hallucinations across both pixel-space diffusion (DDPM) and latent diffusion (LDM). On Hands-11K, VSR reduces hallucinations by $\geq 17\%$ relative to the DDPM baseline, while on *Cards*, reductions exceed 15%. On MNIST, VSR outperforms AAM Oorloff et al. (2025) with at least 1% fewer hallucinations. Across all datasets, VSR achieves the best performance, improving both fidelity (FID, C-FID) and novelty/diversity (FLD). Even in high-dimensional domains such as *ChessImages*, where $p_{\text{train}} \ll p_{\text{data}}$, VSR maintains fidelity while substantially lowering hallucinations (Tab. 5). FLD improvements further indicate that VSR promotes generation of novel, valid samples, a desirable byproduct.

## 7.1 Generalization vs Memorization in ChessImages Dataset

From 190K generated boards, 52% ($\approx$98.8K) are spurious and discarded, leaving 48% ($\approx$91.2K) valid boards. Among these, 10% ($\approx$9.1K) exactly replicate training positions (memorized), while 90% ($\approx$82.1K) represent novel valid boards (generalized). As shown in fig. 2, memorized samples primarily replicate starting formations, whereas generalized samples display more diverse structures.

**Diversity of Generated Boards by Game Phase:** We categorize valid boards into *Opening*, *Midgame*, and *Endgame* using material sum based on standard piece values (pawn=1, knight=3, bishop=3, rook=5, queen=9; kings excluded) Shannon (1950); Smith (1997). With $M_{\text{start}} = 78$, we define phases as: Opening $M \geq 40$, Midgame $20 \leq M < 40$, Endgame $M < 20$. Among 82.1 K novel positions (Fig. 2), this yields $\sim$69.3 K opening (84.4%), 7.5 K midgame (9.1%), and 2.0 K endgame (2.4%) boards, providing a stage-diverse set for chess engine augmentation Silver et al. (2017); Schrittwieser et al. (2020).

| Method | C-FID ↓ | FID ↓ | FLD ↓ | H% ↓ |
|---|---|---|---|---|
| DDPM | **3.74** | 191.68 | 96.83 | 71.00 |
| + VSR $\mathcal{L}_{\text{smooth}}$ | 4.32 | **191.19** | **48.75** | **56.01** |
| LDM-UC | 3.59 | **29.15** | 89.65 | 11.66 |
| + VSR $\mathcal{L}_{\text{smooth}}$ | **3.54** | 34.67 | **52.17** | **9.28** |

Table 5: Chess dataset comparison. VSR reduces hallucinations (H%) while improving novelty (FLD) and maintaining fidelity. Metrics: C–FID (CLIP FID), FID (Inception FID), FLD (Fréchet Latent Distance).

**Invalid Boards** As shown in Fig. 2, any generated configuration that violates the basic piece-count constraints of chess is classified as an invalid board. Our examples in Fig. 2 include cases where one

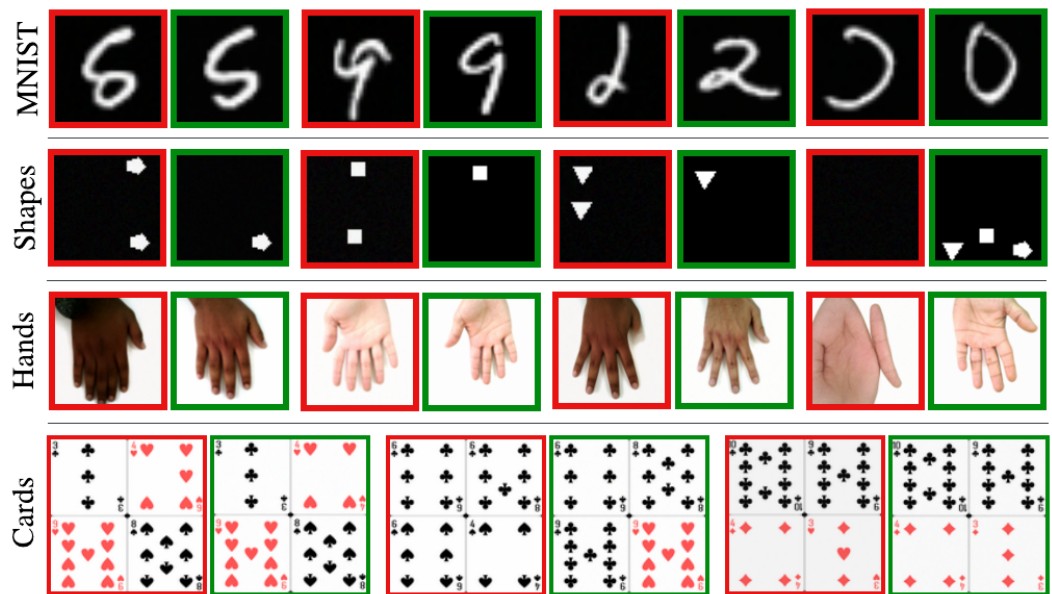

Figure 1: Qualitative examples of corrected hallucinations in DDPM Ho et al. (2020b) with VSR $\mathcal{L}_{\mathrm{smooth}}$. Each pair shows hallucinated (red) vs. non-hallucinated (green) across datasets.

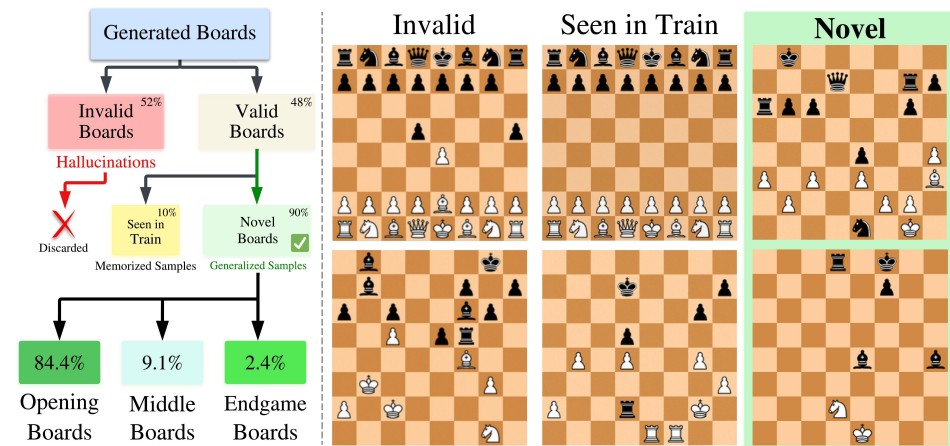

Figure 2: Categorization of generated chessboards into spurious (Invalid), memorized (Seen in Train), and generalized (Novel) samples, with qualitative examples.

side has more than eight pawns (e.g., nine pawns) or three kings appear on the board. Please refer to the Appendix F for more samples of all cases.

## 8 CONCLUSION

We present a density-based perspective on hallucinations in diffusion models, showing that score smoothness induces a nonzero lower bound on probability mass allocated to regions outside the true data support, thereby resulting in a positive hallucination rate. Building on this insight, we propose *Variance-Guided Score Regularization* (VSR), an architecture-agnostic regularizer that increases the score Jacobian to reduce the gap mass lower bound. Empirical results across synthetic, real-world, and newly proposed challenge benchmarks demonstrate that VSR reduces hallucinations by up to $25\%$ while preserving high fidelity and diversity. Beyond a practical mitigation strategy, our work provides theoretical grounding for studying hallucinations in generative diffusion models and contributes new benchmarks for systematic evaluation of hallucinations. **Limitations:** Our approach focuses on reducing, rather than fully eliminating, hallucinations. A systematic understanding of strategies that promote the generation of generalized samples, particularly in natural image domains, remains an open area for future work.

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

Appendix

## A  Towards Zero Hallucinations during generation

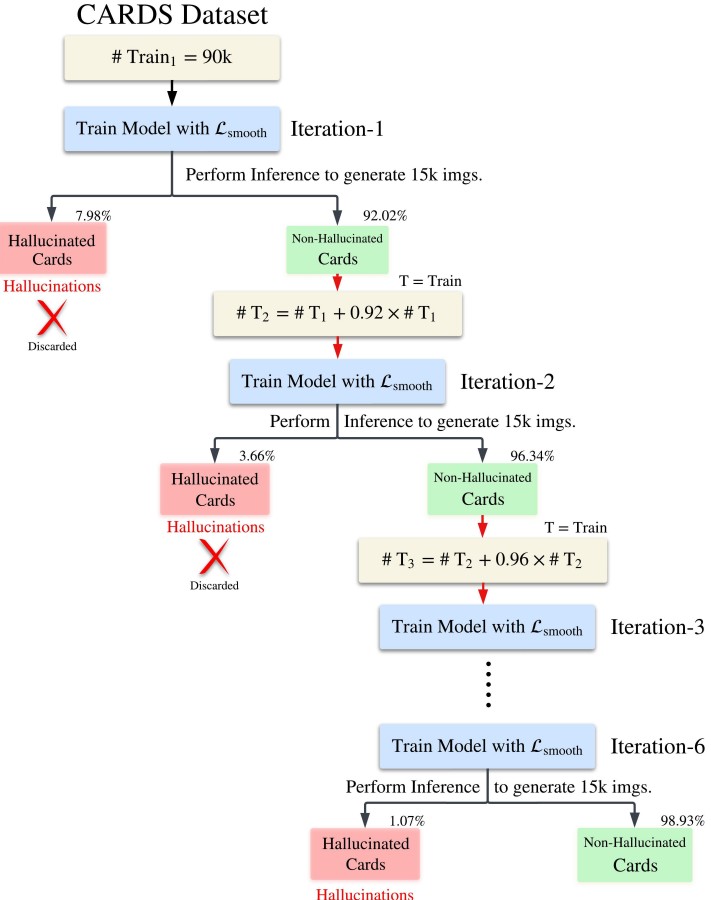

Figure 3: Iterative Training while appending Non-Hallucinated Images to $\mathcal{P}_{\text{train}}$

We propose a way that drives the hallucination rate toward zero. Figure 3 illustrates the effectiveness of our proposed method ($\mathcal{L}_{\text{smooth}}$ loss) within an iterative training strategy to systematically reduce hallucinations during image generation. Beginning with a base model trained on an initial dataset of 90K images, each iteration involves generating 15,000 new card images, filtering out hallucinated samples, and appending only valid, non-hallucinated cards to the training set for the next iteration. This progressively refined dataset, denoted as $p_{\text{train}}$, is then used to retrain the model again from scratch. As shown, hallucination rates drop sharply from 7.98% in iteration-1 to 1.07% by iteration-6, while the proportion of non-hallucinated outputs steadily increases to 98.93%. This iterative bootstrapping approach demonstrates how $\mathcal{L}_{\text{smooth}}$ enables the model to internalize valid patterns and avoid degenerate generations over time, leading to near-zero hallucinations during generation.

| Iteration | Hal. Rate (%) |
|---|---|
| Iteration-1 | 7.98 |
| Iteration-2 | 3.66 |
| Iteration-3 | 2.74 |
| Iteration-4 | 1.82 |
| Iteration-5 | 1.19 |
| Iteration-6 | 1.07 |

Table 6: Reduction in hallucination rate over training iterations using the proposed $\mathcal{L}_{\text{smooth}}$ objective. As iterative training progresses, the rate of hallucinated generations decreases substantially, validating a way towards zero hallucinations.

## B   SCORE DIFFERENCE CORRELATES WITH HALLUCINATIONS

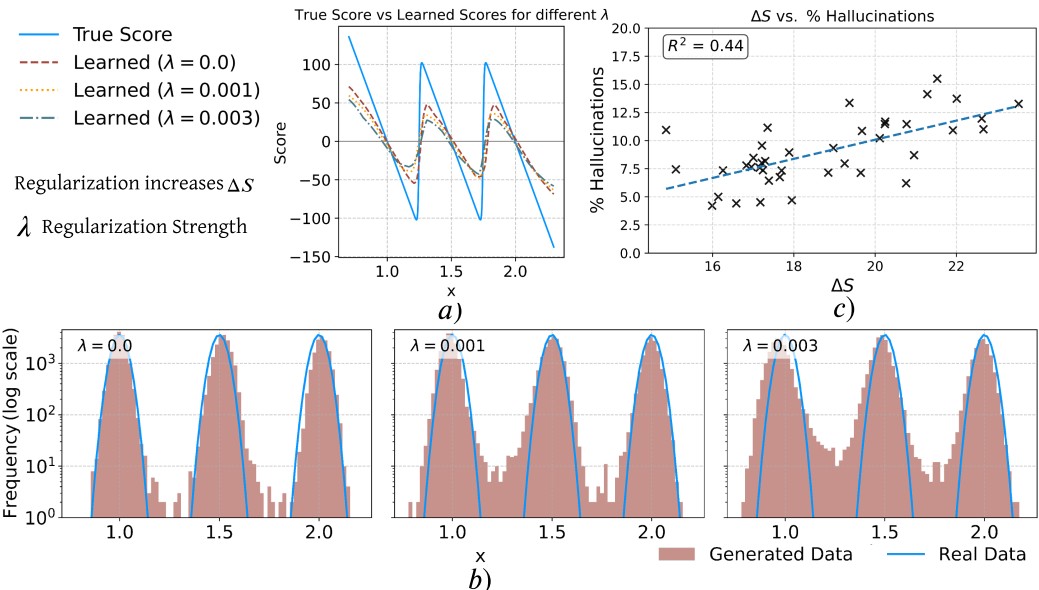

Figure 4: Score smoothing causes hallucinations on 1D Gaussians with means $\mu = [1.0, 1.5, 2.0]$ and $\sigma = 0.035$. a) L2 smoothens the learned score function increasing $\Delta S$. b) The smoothed score function leads to hallucinations. Seen as a mode interpolation outside the support of the true data. c) Increase in Score difference $\Delta S$ positively correlates with Hallucinations on Hands dataset.

To see the effect of the score smoothing, we add L2 regularization to the NN trained to predict the added noise in 1D Gaussian data (Note that this regularization is not same as $\mathcal{L}_{smooth}$ regularization strength $\rho$ which is VSR regularizer proposed for mitigating hallucinations), demonstrated in Fig. 4. GT score is obtained from closed form PDF: $S_{GT}(x_t) = \frac{\sum_{m=1}^{M} -\frac{x_t - \mu_m}{\sigma^2} \exp\left(-\frac{(x_t - \mu_m)^2}{2\sigma^2}\right)}{\sum_{m=1}^{M} \exp\left(-\frac{(x_t - \mu_m)^2}{2\sigma^2}\right)}$. We observe that increasing regularization smoothens learned score more as seen in Fig. 4a. This also increases the number of hallucinations between two data modes as seen in Fig. 4b (outside the support of the original data given by the blue line). We also observe that the number of hallucinations is directly proportional to the score difference $\Delta S$ in the Hands dataset, as demonstrated in Fig. 4c. We calculate $\Delta S$ as described in the Section 3.1. This also motivates us to manipulate the learned score function to address the Hallucinations directly.

## C   SCORE UN-SMOOTHING WITH $\mathcal{L}_{smooth}$

We optimize the model parameters with the objective

$$\mathcal{L}(\theta) = \mathcal{L}_{\text{DSM}}(\theta) + \rho \, \mathcal{L}_{\text{smooth}}(\theta). \tag{3}$$

For all datasets, denoising score matching (DSM) is treated as the primary objective and we choose $\rho$ to be relatively small so that $\mathcal{L}_{\text{DSM}}$ dominates the training signal. Concretely, we set $\lambda = 0.01$ (i.e., the regularizer has one 100th weight of DSM) in our experiments on 1D, with small ranges on all the Images datasets (See appendix D). This ensures that the learned score field remains well aligned with the ground-truth score as optimized by by DSM loss, and that any changes in hallucination behavior are not simply due to score errors.

In fig. 5 we visualize the effect of the regularizer on a 1D Gaussians with means $\mu = [1.0, 1.5, 2.0]$ and $\sigma = 0.035$. The dark-blue curve corresponds to a model trained with DSM only ($\rho = 0$), and the orange dashed curve corresponds to the regularized model ($\rho = 0.01$). VSR regularized model matches the true score in high-density regions around each mode, indicating that the mean behavior and density of in-support samples are preserved. The key difference appears in the low-density gaps:

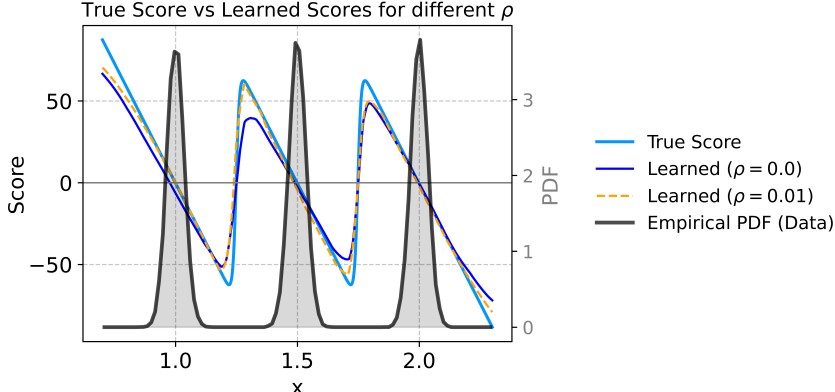

Figure 5: Ablation of regularization strength $\rho$

the regularized model exhibits sharper score transitions and reduced spurious "flattening" between modes, bringing it closer to the piecewise-linear ground-truth score.

This behavior is consistent with the theoretical picture of hallucinations in diffusion models proposed by Aithal et al. (2024), who attribute hallucinations to overly smooth approximations of a ground-truth score that has sharp changes between disjoint high-density regions. In that view, increasing score sharpness in gap regions reduces mode interpolation and lowers hallucination rates. Our regularizer biases the model toward this sharper, more faithful score field while DSM keeps the overall score error low. Together, these properties explain why we observe reduced hallucinations in practice even though the regularizer is applied globally across the domain.

$$\text{H\%} = 100 \times \frac{\#\{\tilde{x}_i : D(\tilde{x}_i) = 1\}}{N}$$

## D    SELECTING REGULARIZATION STRENGTH $\rho$

The fig. 6 shows, there is an optimal $\rho$ at which hallucinations reach their minimum on pixel-space DDPM Ho et al. (2020a). Raising $\rho$ amplifies the $\mathcal{L}_{smooth}$ penalty, shrinking the score difference $\Delta S$ and driving down hallucination rates. However, once $\rho$ exceeds some threshold, the denosing score-matching loss $\mathcal{L}_{DSM}$ becomes under-weighted in comparison with $\mathcal{L}_{smooth}$ and hallucinations begin to increase again. All the image datasets use $\rho = 0.1$ as a weight for $\mathcal{L}_{smooth}$ regularization strength.

## E    DETAILS ON IMPLEMENTATION OF $\mathcal{L}_{smooth}$

As seen in Section 3.1, $L_{\text{smooth}}$ penalizes small Jacobians of the score function. For data in $\mathbb{R}^D$, the Jacobian of the score $S : \mathbb{R}^D \to \mathbb{R}^D$ is a $D \times D$ matrix. In practice, we replace the exact derivatives with a centered finite-difference approximation.

1D CASE

When $D = 1$, $S : \mathbb{R} \to \mathbb{R}$, the Jacobian reduces to the scalar derivative

$$J_S(x) \;=\; \frac{d}{dx}S(x) \;\approx\; \frac{S(x+\varepsilon) \;-\; S(x-\varepsilon)}{2\,\varepsilon}.$$

2D CASE

When $D = 2$, write $x = (x_1, x_2) \in \mathbb{R}^2$ and $S = (S_1, S_2)$. The Jacobian matrix $J_S(x)$ has entries

$$[J_S(x)]_{ij} \;=\; \frac{\partial S_i(x)}{\partial x_j} \;\approx\; \frac{S_i\big(x+\varepsilon\,e_j\big) \;-\; S_i\big(x-\varepsilon\,e_j\big)}{2\,\varepsilon} \quad (i,j=1,2),$$

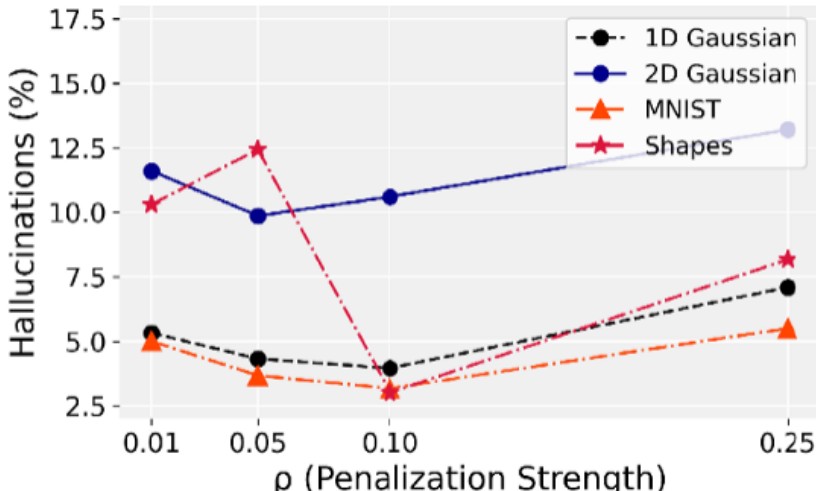

Figure 6: Ablation of regularization strength $\rho$

where $e_1 = (1, 0)$, $e_2 = (0, 1)$. Equivalently,

$$J_S(x) \approx \frac{1}{2\varepsilon} \begin{pmatrix} S_1(x_1 + \varepsilon, x_2) - S_1(x_1 - \varepsilon, x_2) & S_1(x_1, x_2 + \varepsilon) - S_1(x_1, x_2 - \varepsilon) \\ S_2(x_1 + \varepsilon, x_2) - S_2(x_1 - \varepsilon, x_2) & S_2(x_1, x_2 + \varepsilon) - S_2(x_1, x_2 - \varepsilon) \end{pmatrix}.$$

**Images case**

The Jacobian of the Score is also the Precision matrix $-\Sigma^{-1}$. However, there are two problems in calculating the covariance matrix $\Sigma$, 1. Closed-form PDF is not available 2. Calculating and storing the Jacobian is not computationally feasible for high-dimensional image settings. Therefore, instead, we use the $\Sigma_\phi$ learned in the denoising process. We adopt the DDPM-IP parameterization Nichol & Dhariwal (2021) to learn variance, more details in Section 3.1 of the main paper.

## F   MORE DETAILS ON THE CHESSIMAGES DATASET

**Invalid Chessboard Detection:**   Section 5.1 in the main paper describes details about creating the ChessImages dataset. The validation module ensures that every generated chessboard image corresponds to a legal board configuration. This is achieved through a hybrid pipeline comprising visual and rule-based checks, designed to detect hallucinations automatically—board states violating chess semantics or displaying visual inconsistencies. We begin by reconstructing the FEN string from each rendered image using a template-matching-based parser, achieving 100% reconstruction accuracy on the training set. However, given chess's combinatorial nature, no tractable algorithm can verify the reachability of arbitrary board states through legal move sequences. Hence, we instead focus on verifying the structural validity of the board state using syntactic and semantic criteria.For the validation of the FEN, we use `status()` from the `python-chess` library[5]. We construct a dataset with strong structural priors and verifiable correctness by enforcing these constraints. This eliminates manual annotation during hallucination detection and enables reproducible and objective evaluations in downstream generative modeling tasks.

A generated chessboard image is considered invalid if it meets any of the following criteria: (1) the extracted FEN string from the image has a similarity score below 50%, indicating poor or ambiguous visual parsing; or (2) the parsed FEN fails legality checks using the python-chess library, such as having multiple kings of the same color, exceeding eight pawns per player, overlapping or missing pieces, or violating fundamental chess rules.

Below we list the rules used by the chess library's `status()` check and, for each rule, we also show the images flagged as "hallucinated" because of violations of these rules in Fig. 7 and Fig. 8.

---

[5] https://python-chess.readthedocs.io/en/latest/

(i) **Non-empty board.** A valid FEN must contain at least one piece. Violations are flagged by `STATUS_EMPTY`.

(ii) **Exactly one white king.** There must be one (and only one) white king on the board. Violations are flagged by `STATUS_NO_WHITE_KING`, `STATUS_TOO_MANY_KINGS`.

(iii) **Exactly one black king.** There must be one (and only one) black king. Violations are flagged by `STATUS_NO_BLACK_KING`, `STATUS_TOO_MANY_KINGS`.

(iv) **Piece-count limits.** No side may have more than 16 pieces. Violations are flagged by `STATUS_TOO_MANY_WHITE_PIECES`, `STATUS_TOO_MANY_BLACK_PIECES`.

(v) **Pawn-count limits.** No side may have more than eight pawns. Violations are flagged by `STATUS_TOO_MANY_WHITE_PAWNS`, `STATUS_TOO_MANY_BLACK_PAWNS`.

(vi) **No pawns on back-rank.** Pawns may not appear on ranks 1 or 8. Violations are flagged by `STATUS_PAWNS_ON_BACKRANK`.

(vii) **Legal castling rights.** Castling flags must match the king/rook placement. Violations are flagged by `STATUS_BAD_CASTLING_RIGHTS`.

(viii) **Valid en passant.** The ep-target square must be reachable by a two-square pawn move. Violations are flagged by `STATUS_INVALID_EP_SQUARE`.

(ix) **No opposite-side check.** The side that is not to move cannot be checked. Violations are flagged by `STATUS_OPPOSITE_CHECK`.

(x) **Max two checking pieces.** At most two pieces may deliver a check. Violations are flagged by `STATUS_TOO_MANY_CHECKERS`.

(xi) **Possible check sequence.** Checks must arise via a legal move (including ep pushes). Violations are flagged by `STATUS_IMPOSSIBLE_CHECK`.

A Standard FEN string contains `"<PiecePlacement> <ActiveColor> <CastlingRights> <EnPassant> <HalfmoveClock> <FullmoveNumber>"`. Template matching can only give us `<PiecePlacement>`. Therefore, the rules (vii, viii, ix) that use the information from parts of the FEN other than `<PiecePlacement>` are ignored in our work.

Demonstrated in Tab. 7 we compare the total number of valid novel boards generated by all the methods. Proposed methods can be utilized as more robust data augmentation technics with high rule prior datasets such as proposed ChessImages dataset.

| Method | # Novel Boards |
|---|---|
| DDPM Ho et al. (2020b) | 60842 |
| Variance Learning Nichol & Dhariwal (2021) | 69950 |
| $L_{\text{smooth}}$ | 77421 |

Table 7: Number of valid boards that are novel out of 190K generated samples. Both Variance learning and $L_{\text{smooth}}$ gnerate considerably large number of valid novel boards as compared to baselines.

## G    EFFECT OF DATASET SIZE

We investigate how the size of the training set influences hallucination rates. To ensure consistent comparisons, we construct three nested subsets containing 75%, 50%, and 25% of the full dataset—each smaller subset being wholly contained within the next larger one. As shown in Tab. 8, shrinking the training set reduces the support from diverse examples, which in turn increases the incidence of hallucinations. This underscores the crucial role of ample data support in achieving reliable image generation.

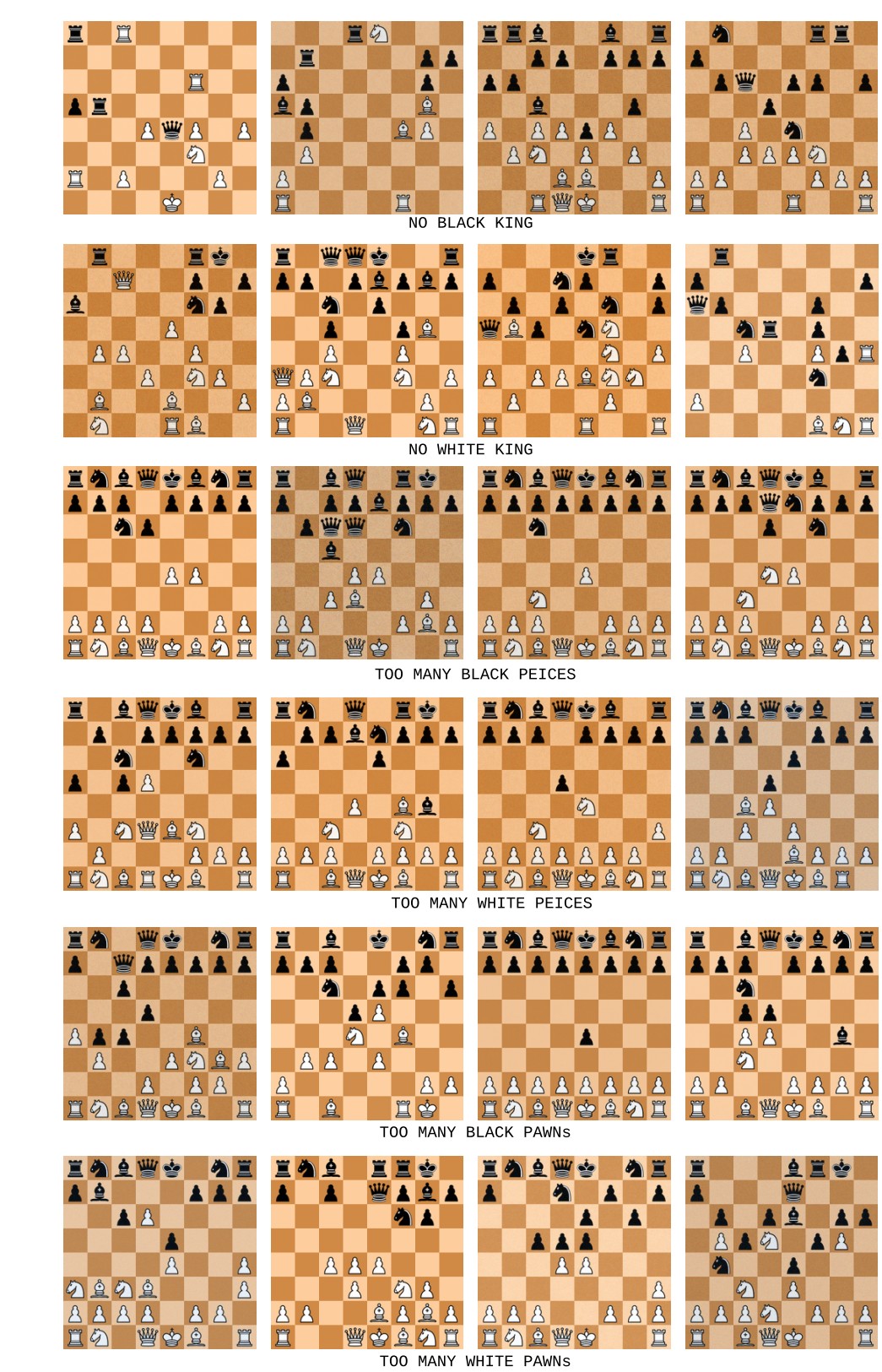

Figure 7: Generated images marked Hallucinated for the reasons mentioned at the bottom of each row. Although the image brightness varies, simple ad-hoc techniques can readily correct it. Our primary focus, however, is on preserving the semantic details of the ChessImages dataset.

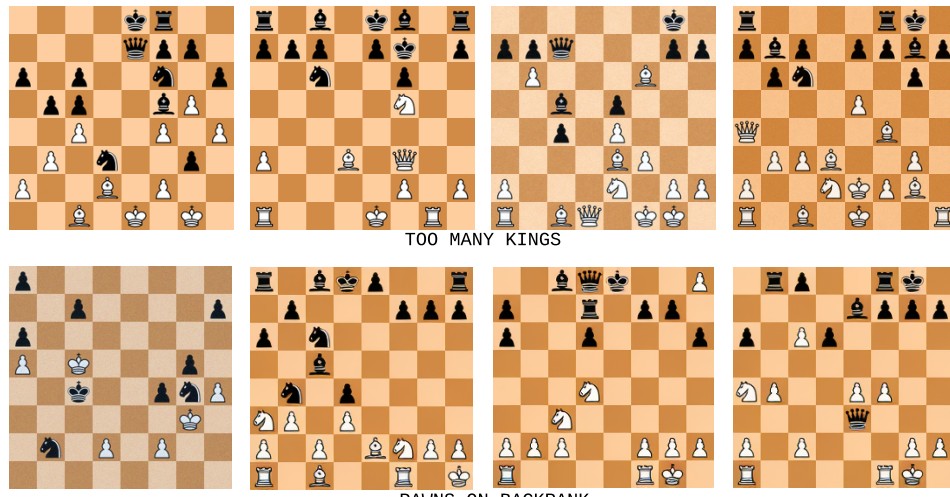

Figure 8: Generated images marked Hallucinated for the reasons mentioned at the bottom of each row. Although the image brightness varies, simple ad-hoc techniques can readily correct it. Our primary focus, however, is on preserving the semantic details of the ChessImages dataset.

| Denoising Steps | Hallucinations (%) | |
|---|---|---|
| | DDPM | $L_{\text{smooth}}$ |
| 50 | 61.75 | 57.75 |
| 100 | 62.00 | 53.00 |
| 150 | 69.50 | 55.75 |
| 200 | 64.00 | 51.00 |
| 250 | 66.25 | 55.00 |

Table 9: Effect of denoising steps on hallucinations (%) on the Chess dataset.

| Dataset Size | Shapes | | ChessImages |
|---|---|---|---|
| | DDPM | $L_{\text{smooth}}$ | $L_{\text{smooth}}$ |
| 25 | 89.74 | 20.50 | 80.33 |
| 50 | 57.16 | 13.16 | 65.35 |
| 75 | 55.16 | 5.66 | 61.75 |
| 100 | 29.50 | 3.00 | 55.0 |

Table 8: Effect of training-set size on hallucination rates (%): for the Shapes dataset we compare DDPM vs. $L_{\text{smooth}}$, and for ChessImages we report on $L_{\text{smooth}}$.

## H    EFFECT OF NUMBER OF DENOISING STEPS ON HALLUCINATIONS

We investigate how varying the number of denoising steps during inference affects hallucination rate on the Chess dataset. As table 9 shows, there is no discernible relationship between step count and hallucination rate. Although fewer denoising steps are known to degrade overall image fidelity, they do not consistently alter the number of hallucinations.

## I    LDM ROMBACH ET AL. (2022) PROMPT TUNING

For the conditional LDM (LDM-C) setting, we condition generation on text prompts: a single default prompt for the Hands dataset, and class-embedded prompts for MNIST. In the prompt-tuning (LDM-PT) setting, we further fine-tune these prompts to mitigate the hallucinations we observed (see Table 3 in the main paper). For each dataset, we crafted 20 distinct prompts and, at inference

time, randomly select one to drive image synthesis. We observe that this prompt-tuning strategy substantially reduces hallucination rates on both Hands and MNIST.

**MNIST:**

Default Prompt:

```
["Image of handwritten digit <digit_class>"]
```

Finetuned Prompts:

```
[ # I. Zero
"MNIST-style handwritten 'zero': thin white strokes, centered on a
    clean black background, no extra marks.",
"MNIST-style handwritten 'zero': minimal white loop, centered on
    black, uniform thickness, no noise.",

# II. One
"MNIST-style handwritten 'one': single thin white vertical stroke,
    centered on black, no stray pixels.",
"MNIST-style handwritten 'one': clean white digit one, straight line,
    centered on black, isolated.",

# III. Two
"MNIST-style handwritten 'two': crisp white strokes, centered on
    black, no overlapping or smudges.",
"MNIST-style handwritten 'two': clear white digit two, centered on
    black, uniform lines, no noise.",

# IV. Three
"MNIST-style handwritten 'three': two smooth thin white strokes,
    centered on black, no extra artifacts.",
"MNIST-style handwritten 'three': neat white digit three, centered on
    black, distinct curves, clean.",

# V. Four
"MNIST-style handwritten 'four': intersecting thin white strokes,
    centered on black, no stray marks.",
"MNIST-style handwritten 'four': crisp white digit four, centered on
    black, clear junctions.",

# VI. Five
"MNIST-style handwritten 'five': clear thin white strokes, centered
    on black, no overlapping lines.",
"MNIST-style handwritten 'five': sharp white digit five, centered on
    black, isolated strokes.",

# VII. Six
"MNIST-style handwritten 'six': continuous thin white stroke,
    centered on black, no breaks.",
"MNIST-style handwritten 'six': clean white digit six, rounded form,
    centered on black, no noise.",

# VIII. Seven
"MNIST-style handwritten 'seven': two thin white strokes, centered on
    black, no extra marks.",
"MNIST-style handwritten 'seven': neat white digit seven, centered on
    black, uniform thickness.",

# IX. Eight
"MNIST-style handwritten 'eight': two distinct thin white loops,
    centered on black, no distortions.",
"MNIST-style handwritten 'eight': symmetric white digit eight,
    centered on black, clear separation.",
```

```
    # X. Nine
    "MNIST-style handwritten 'nine': thin white strokes, centered on
        black, isolated and clean.",
    "MNIST-style handwritten 'nine': crisp white digit nine, centered on
        black, no extra pixels."]
```

**Hands:**

Default Prompt:

```
    ["Close-up high quality image of a human hand on White background"]
```

Finetuned Prompts:

```
    ["High-resolution photo of a human hand, palm fully open with five
        fingers (thumb, index, middle, ring, pinky) spread naturally,
        plain white background.",

    "Close-up shot of an open human palm showing all five fingers in
        correct thumb-to-pinky order, flat facing the camera, on white.",

    "Photograph of a human hand with palm wide open, five straight
        fingers (thumb - index - middle - ring - little finger), against
        a white backdrop.",

    "Studio image of an open palm displaying five fingers in proper
        sequence-thumb at left, pinky at right-on a clean white
        background.",

    "Realistic photo of a single human palm, five fingers fully extended
        in thumb-to-pinky order, flat and facing forward, white
        background.",

    "High-quality image of a human hand, palm completely open, five
        fingers aligned anatomically (thumb, index, middle, ring, pinky),
         white backdrop.",

    "Close-up of an open palm with five straight fingers, thumb on the
        left and pinky on the right, on solid white.",

    "Photorealistic shot of a fully opened palm showing five fingers in
        correct order, flat against a white background.",

    "Sharp photo of a human hand, palm fully extended with thumb, index,
        middle, ring, and little finger visible in order, white
        background.",

    "Clean studio portrait of an open palm-five fingers (thumb through
        pinky) splayed evenly-on a white backdrop.",

    "High-resolution image of an open palm with five anatomically ordered
         fingers, thumb first then index, middle, ring, and pinky,
        against white.",

    "Close-up studio photo of a human palm fully open, showing five
        straight fingers in thumb-to-pinky sequence, white background.",

    "Real-life shot of an open hand with palm facing camera, five fingers
         (thumb - index - middle - ring - little) in order, white
        backdrop.",

    "Crisp image of an open palm with five fingers aligned anatomically,
        thumb on the left edge, pinky on the right, plain white
        background.",
```

```
    "Photograph of a human palm flat and facing forward, five fingers
        visible in correct anatomical order, white background.",

    "Studio-style image of an open hand-five fingers from thumb to pinky-
        fully extended and flat against white.",

    "Close-up of a human palm with five distinct fingers, starting from
        thumb then index, middle, ring, little, on a white backdrop.",

    "Detailed photo of an open palm showing five fingers in sequence,
        thumb at outer edge, pinky at other, on solid white.",

    "High-detail shot of a human palm fully opened, five straight fingers
         in anatomical order, flat and white background.",

    "Clear photo of a human hand, palm fully open with thumb, index,
        middle, ring, and pinky fingers visible in order on a white
        background."]
```

## J  MORE DETAILS ON THE CARDS DATASET

In 9 we show more samples of the images that are hallucinated by the rules mentioned in the main paper.

## K  IMPLEMENTATION DETAILS

For 1D and 2D datasets, our code is built upon Aithal et al. (2024). For Image datasets with variance learning and $L_{smooth}$ implementation, we build upon Nichol & Dhariwal (2021). All experiments are carried out on 8 Nvidia A6000 GPUs. All the quantitative results on the Image datasets are obtained using six seeds and generating 100 images per seed. We also used six seeds for the 1D and 2D cases, generated 1 million sample points per seed, and reported the average. For the LDM baseline, we use the codebase provided by Rombach et al. (2022). Specifically, for LDM-C, we initialized our diffusion model from the Stable Diffusion checkpoint pretrained on ImageNet and used the CLIP text encoder to extract text embeddings. For unconditional training, we train LDM from scratch. For Oorloff et al. (2025), we directly use the quantitative results reported in the original paper.

## L  HALLUCINATION REDUCTION IN REAL WORLD CATS-VS-DOGS DATASET

In addition to the five datasets (synthetic and real-world datasets) we also present qualitative results on real-world cats and dogs image dataset in fig. 11 to show the effectiveness of our proposed method. We identify that using $\mathcal{L}_{\text{smooth}}$ leads to significantly reduced hallucination rate.

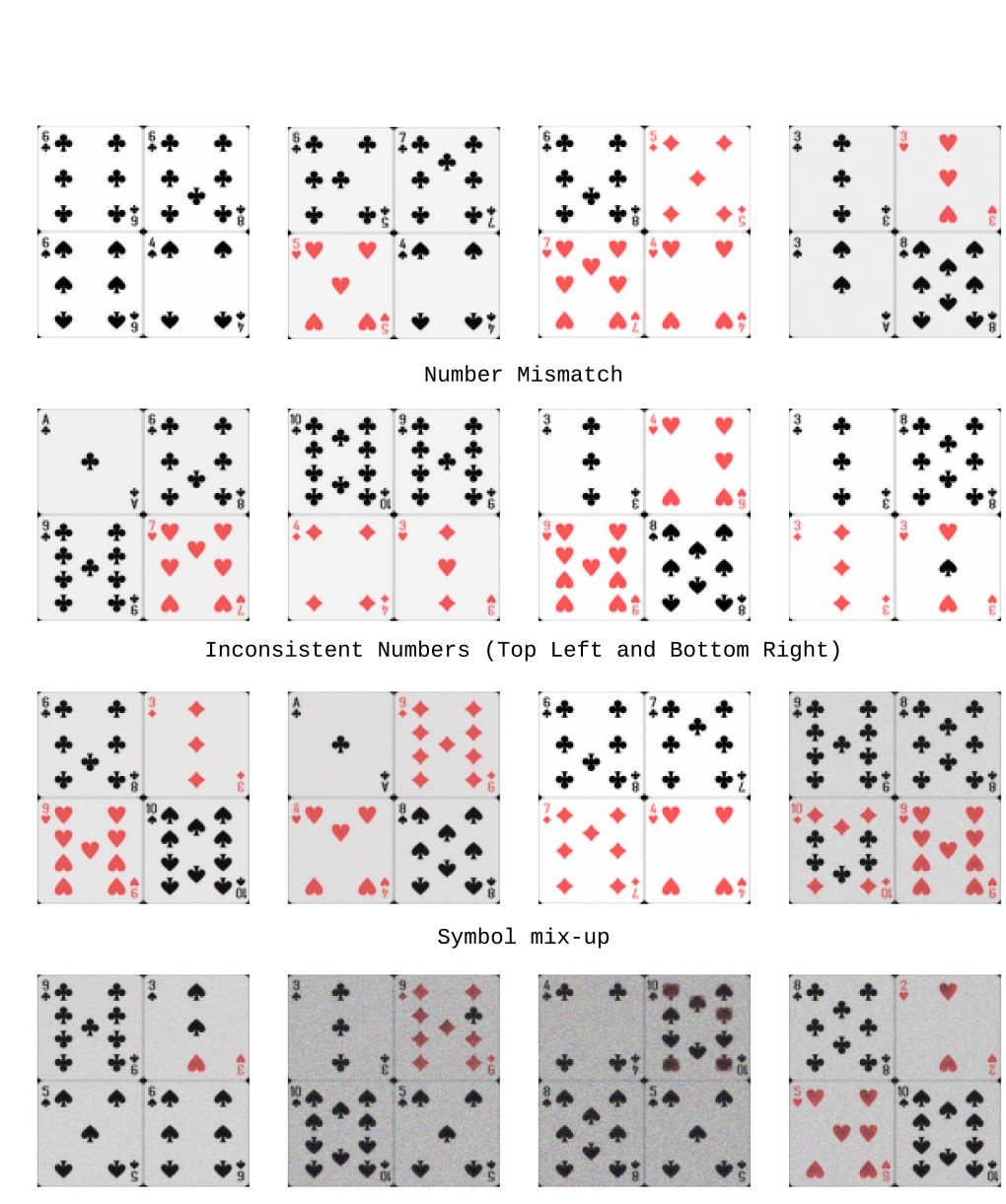

Figure 9: Generated images marked Hallucinated for the reasons mentioned at the bottom of each row. Although the image brightness varies, simple ad-hoc techniques can readily correct it. Our primary focus, however, is on preserving the semantic details of the Cards dataset

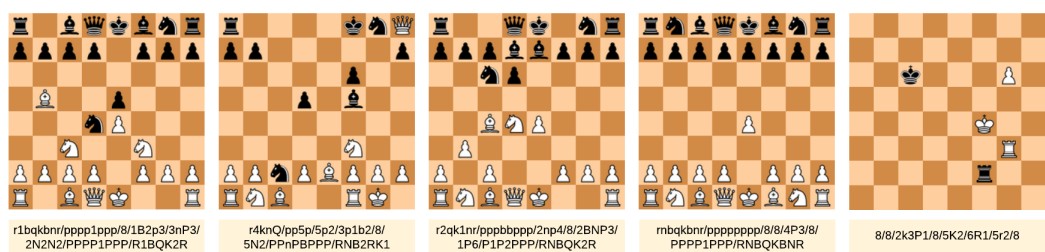

r1bqkbnr/pppp1ppp/8/1B2p3/3nP3/2N2N2/PPPP1PPP/R1BQK2R

r4knQ/pp5p/5p2/3p1b2/8/5N2/PPnPBPPP/RNB2RK1

r2qk1nr/pppbbppp/2np4/8/2BNP3/1P6/P1P2PPP/RNBQK2R

rnbqkbnr/pppppppp/8/8/4P3/8/PPPP1PPP/RNBQKBNR

8/8/2k3P1/8/5K2/6R1/5r2/8

Figure 10: Example samples from the proposed *ChessImages* dataset. *Top*: a generated chessboard configuration. *Bottom*: its corresponding Forsyth–Edwards Notation (FEN) string, providing an exact symbolic representation of the board state.

Hallucinated   Non-Hallucinated   Hallucinated   Non-Hallucinated   Hallucinated   Non-Hallucinated

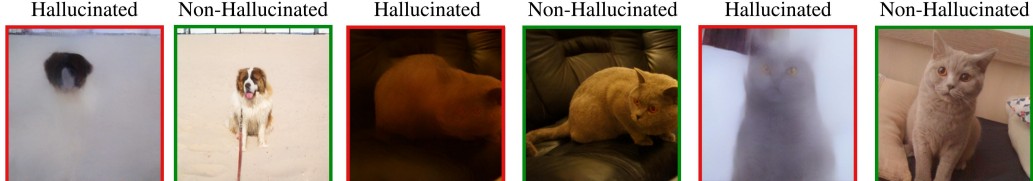

Figure 11: VSR corrects hallucinations over LDM Baseline.