# OpenReview forum: "Variance-Guided Score Regularization for Hallucination Mitigation in Diffusion Models"
_ICLR.cc/2026/Conference — Submitted to ICLR 2026_

### Official Review · Reviewer_RihT · 2025-10-30

**Soundness:** 2
**Presentation:** 3
**Contribution:** 2
**Rating:** 4
**Confidence:** 4

**Summary:**

This paper provides a theoretical and practical framework for mitigating hallucinations in diffusion models. The authors start by offering a density-based formulation of hallucinations, showing that the Gaussian nature of the diffusion process inevitably assigns non-zero probability to regions outside the true data support. They derive an explicit lower bound on this off-support mass as a function of the score function’s Lipschitz constant and boundedness, establishing a link between score smoothness and hallucination probability.
To mitigate hallucinations, they propose Variance-Guided Score Regularization (VSR) — a training-time regularizer that penalizes low Jacobian norms of the score function, effectively tightening the lower bound on off-support probability. The method is architecture-agnostic and leverages variance information learned during denoising. Empirical results on both standard datasets (MNIST, Hands, Shapes) and two new benchmarks (Cards and ChessImages) demonstrate consistent reductions in hallucinations (15–25%) while maintaining fidelity and diversity.

**Strengths:**

1. The paper is the first to explicitly connect score smoothness in diffusion models to hallucination probability, using measure-theoretic reasoning and formal lower bounds.

2. The smoothness penalty is a mathematically motivated modification that directly influences the score Jacobian. Its integration with variance estimation (via I-DDPM) adds interpretability and practical feasibility.

3. The authors conduct experiments on various datasets, showing reduction in both hallucination rate (H%) and score error (∆S).

**Weaknesses:**

1. The theoretical foundation of VSR hinges on a simplified assumption that disjoint supports (gaps) exist in high-dimensional data manifolds. This assumption holds for synthetic or low-dimensional distributions (e.g., 1D/2D Gaussians, discrete combinatorial datasets like Cards/ChessImages), but it may easily break down for continuous, correlated natural image manifolds. For example, in the high-dimensional image space, distributions of semantically different classes such as cats vs. dogs, or cats vs. tigers, may overlap in the feature manifold as they share visual attributes like eyes, fur, and texture.

2. Lemma 1, lacking a formal proof and relying on the above assumption, is more a heuristic hypothesis than a mathematically valid lemma. As a result, the derived bounds may not apply to real-world generative tasks where class manifolds are continuous and overlapping.

3. The lower bound derived in Theorem 2 relies on strong regularity assumptions including L-Lipschitz continuity, bounded scores, and compact boundaries, which rarely hold for high-dimensional natural images. The bounds thus provide conceptual intuition but limited quantitative predictive power.

4. The proposed regularizer encourages larger Jacobians, but intuitively, large Jacobians can also cause instability or excessive curvature in score fields, potentially harming training stability if not carefully tuned. This tension between theory and practice is not thoroughly discussed.

**Questions:**

1. Could the approach generalize to text-conditioned or multi-modal diffusion models, where hallucinations stem from semantic misalignment rather than density leakage?

2. Given that theorem 2 assumes compact boundaries, how does this extend to unbounded data manifolds like real image spaces?

3. Have the authors evaluated how the score smoothness affects mode coverage vs. fidelity trade-off in FID metrics?

---

> ### Author Response · Authors · 2025-12-04
> **Response to Reviewer-RihT**
>
> **Overlap of feature manifold in high-dimensional datasets**
>
> We appreciate this thoughtful concern and agree that the assumption of perfectly disjoint supports is a simplification. Our aim is not to claim that high-dimensional natural image classes like “cats” and “dogs” occupy disjoint manifolds in pixel or feature space, but to isolate a mechanism that recent work [1] has already highlighted: hallucinations in diffusion models arise because the *learned* score is a **smooth approximation** to a ground-truth score that has sharp, almost step-like transitions between regions where the data density is high and regions where it is essentially zero. This smoothing causes the model to interpolate between such regions and assign non-negligible probability to low-density “gap-like” areas; when the true (sharp) score is used at sampling time, these interpolations and hallucinations largely disappear [1]. Our mixture-with-gaps model is a stylized abstraction of this phenomenon. It holds exactly in our synthetic (1D/2D Gaussians) and combinatorial (Cards/ChessImages) settings, where the data distribution *does* have hard structural constraints that induce genuine gaps, and it provides a qualitatively accurate picture for structured hallucinations in natural images (e.g., extra fingers, shapes in impossible regions, illegal chess boards), where the relevant low-density regions are defined by semantic/structural violations rather than broad class labels like “cat vs. dog.” Within this framework, increased **score sharpness** across these effective boundaries is desirable: it makes the learned score closer to the ideal score, reducing interpolation into gap regions. Our variance-guided regularizer is added on top of denoising score matching (DSM) as $\mathcal{L}=\mathcal{L}_{\text{DSM}} + \rho \mathcal{L}_{\text{smooth}}$
>
> with small $\rho$, so DSM continues to anchor the score to the true score where data density is high, while the regularizer provides a mild bias among DSM-consistent solutions, encouraging sharper transitions in low-density areas. As illustrated in Appendix C, compared to $\rho=0$ the regularized model $(\rho>0)$ (i) better matches the true score around high-density modes, preserving in-support behavior, and (ii) makes the transitions between modes sharper and closer to the true score precisely in the gaps where [1] identify smooth approximations as the source of hallucinations. Thus, even though the global “disjoint support” assumption is an idealization for natural images, it captures the right local structure around semantic/structural gaps, and the resulting theory directly informs a regularizer whose empirical behavior on Hands, Shapes, Cards, and ChessImages is consistent with this mechanism.
>
> **Behavior of the regularizer in the gap regions**
> [1] show that hallucinations in diffusion models arise because the learned score provides a *smooth* approximation to a ground-truth score that has sharp “step-like’’ transitions between disjoint modes. This smoothing causes the model to interpolate between modes, assigning non-negligible probability to low-density gap regions. Conversely, when the *true* (sharp) score is used at sampling time, hallucinations essentially disappear [1].
>
> In light of this, increased score sharpness in gaps is *desirable*: it brings the learned score closer to the ideal step function and reduces the interpolation that causes hallucinations. Our regularizer is added on top of denoising score matching (DSM) as
>
> $$
> \mathcal{L} = \mathcal{L}_{DSM} + \rho \mathcal{L}_{smooth}, \rho=0.1
> $$
>
> so DSM still anchors the score to the ground-truth score function, and the regularizer only provides a mild bias among DSM-consistent solutions. We now explicitly report the range of $\rho$ in appendix section D.
>
> Figure in section C in appendix illustrates this effect. Relative to the unregularized model ($\rho=0$, dark blue), the regularized model ($\rho=0.01$, orange) (i) better matches the true score around the high-density modes (the zeros and slopes near the peaks of the empirical PDF), ensuring that in-support samples and their mean behavior are **preserved**, and simultaneously (ii) produces sharper transitions between modes that are closer to the true score in the low-density gaps. This is precisely the regime where [1] identify smooth score approximations as the source of mode interpolation. Thus, our regularizer *increases* sharpness where it matters for hallucination mitigation-boundaries-DSM ensures accurate modeling of the high-density and low-density regions. Together, this explains why we observe reduced hallucination even though the regularizer is applied globally.
>
> [1] https://arxiv.org/abs/2406.09358

---

### Official Review · Reviewer_Uy7M · 2025-10-30

**Soundness:** 2
**Presentation:** 1
**Contribution:** 2
**Rating:** 2
**Confidence:** 4

**Summary:**

The paper studies hallucinations in diffusion models from a density-based perspective and argues that smooth (Lipschitz) score fields inevitably assign non-zero probability mass to “gap” regions outside the true data support. Building on this, the authors derive a pointwise lower bound that connects the local score bound (S) and Lipschitz constant (L) to off-support density. They then propose a Variance-Guided Score Regularization (VSR) term that encourages larger score Jacobians during training, aiming to reduce the theoretical lower bound of gap mass. Empirically, the paper reports hallucination reductions on synthetic 1D/2D Gaussians and several image datasets, and introduces two structured benchmarks—Cards and ChessImages.

**Strengths:**

1. Foundational problem. Hallucination in diffusion models is a low-level, system-level reliability issue; giving it a precise formalization and trying to tie it to the geometry/smoothness of the score field is valuable.
2. Theory + benchmarks. The paper provides an explicit lower-bound analysis (linking score smoothness to off-support mass) and contributes two large, structured evaluation suites (Cards, ChessImages) with efficient detectors, which can help standardize hallucination studies.

**Weaknesses:**

1. The introduction is a bit misleading. It mentions hallucinations in large T2I models like FLUX and Stable Diffusion 3.5 (L31–35, L39-42) and links them to fairness issues, but the proposed score regularization method can’t directly be applied to those models. These models are flow-matching based and don’t expose their variance heads. It would be good to clarify what kind of diffusion models this method can actually work on, and whether it’s limited to toy.

2. The jump from the theorem to the method isn’t convincing. Section 4 argues that smooth scores lead to nonzero mass in “gap” regions, but the proposed loss simply increases the Jacobian norm everywhere. There’s no argument that this increase happens only on the data manifold. In low-density regions the score and Jacobian can already be large, so this might even amplify off-support gradients—the opposite of what’s intended. The paper should either analyze this behavior region-wise or show empirically that larger Jacobians actually correlate with fewer hallucinations.

3. The paper is hard to follow, e.g., Tables 2 and 3 are confusing.
   - Table 2 shows H% values above 100 (e.g., 521.73) even though it’s described as a percentage.  I couldn’t find a clear explanation of what these numbers mean or how they’re computed in the paper.
   - The same metric switches notation between H% and %H in Table 3.

4. Figure 4b is also hard to interpret. The plot suggests that when $\lambda=0.0$, the learned score is closer to the true score, and increasing $\lambda$ makes it worse. This looks opposite to the claim that the regularizer improves the score field. In L252, it seems the regularization strength is the weight of the proposed loss?

5. It’s not clear how the smoothness loss is balanced with the standard denoising loss over time. The score magnitude changes a lot with t, so applying the same weight $\rho$ everywhere might over-regularize some steps. The paper doesn’t show any per-t weighting or sensitivity analysis for $\rho$ and $\eta$. Some discussion on how to tune or schedule this term would make the method easier to reproduce and understand.

**Questions:**

Please refer to Weaknesses section.

---

> ### Author Response · Authors · 2025-12-04
> **Response to Reviewer-Uy7M**
>
> We thank the reviewer for their insightful comments. We address all their concerns below:
>
> **Large T2I Models like FLUX Stable Diffusion 3.5**
>
> We appreciate this observation and agree that our current introduction can be read as suggesting that our regularizer is directly applicable to large proprietary T2I models like FLUX or Stable Diffusion 3.5. Our intent in mentioning those models was to **motivate the practical importance of hallucinations**, not to claim that we apply our method to them. The proposed variance-guided score regularizer is designed for **score-based diffusion models that expose (or can be trained with) a denoising covariance/variance head**, such as DDPM / Improved DDPM and latent diffusion models with a learnable variance parameterization. In these settings we have access to the internal score network and its variance proxy, which we regularize. In contrast, current large T2I systems like FLUX are typically **flow-matching based, closed-source, and do not expose a variance head**, so we cannot apply our method to them out of the box in this work. We will revise the introduction to clearly separate (i) the **motivating examples** of hallucinations in large-scale T2I models from (ii) the **class of models we actually target and evaluate**, and to emphasize that our theoretical analysis is architecture-agnostic while our concrete regularizer is instantiated and tested on accessible score-based diffusion and latent diffusion models (including non-toy image datasets such as Hands, Shapes, Cards, and ChessImages), rather than on proprietary T2I systems.
>
> **The paper should either analyze this behavior region-wise or show empirically that larger Jacobians actually correlate with fewer hallucinations.**
>
> We thank the reviewer for pointing out the connection between our theorem and the behavior of the regularizer. [1] show that hallucinations in diffusion models arise because the learned score provides a *smooth* approximation to a ground-truth score that has sharp “step-like’’ transitions between disjoint modes. This smoothing causes the model to interpolate between modes, assigning non-negligible probability to low-density gap regions. Conversely, when the *true* (sharp) score is used at sampling time, mode interpolation and hallucinations essentially disappear [1].
>
> In light of this, increased score sharpness in gaps is *desirable*: it brings the learned score closer to the ideal step function and reduces the interpolation that causes hallucinations. Our regularizer is added on top of denoising score matching (DSM) as
>
> $$\mathcal{L} = \mathcal{L}_{\text{DSM}} + \rho\,\mathcal{L}_{\text{smooth}},\quad \rho \approx 0.1$$
>
> so DSM still anchors the score to the ground-truth score function, and the regularizer only provides a mild bias among DSM-consistent solutions. We now explicitly report the range of $\rho$ in appendix section D.
>
> Figure in section C in appendix illustrates this effect. Relative to the unregularized model ($\rho=0$, dark blue), the regularized model ($\rho=0.01$, orange) (i) better matches the true score around the high-density modes (the zeros and slopes near the peaks of the empirical PDF), ensuring that in-support samples and their mean behavior are **preserved**, and simultaneously (ii) produces sharper transitions between modes that are closer to the true score in the low-density gaps. This is precisely the regime where [1] identify smooth score approximations as the source of mode interpolation. Thus, our regularizer *increases* sharpness where it matters for hallucination mitigation-boundaries-DSM ensures accurate modeling of the high-density and low-density regions. Together, this explains why we observe reduced hallucination even though the regularizer is applied globally.
>
> [1] Aithal, S.K., Maini, P., Lipton, Z. and Kolter, J.Z., 2024. Understanding hallucinations in diffusion models through mode interpolation. Advances in Neural Information Processing Systems, 37, pp.134614-134644.

---

### Official Review · Reviewer_fxRj · 2025-10-31

**Soundness:** 3
**Presentation:** 4
**Contribution:** 1
**Rating:** 8
**Confidence:** 4

**Summary:**

This paper addresses the problem of hallucinations (implausible samples outside the true data distribution support) in Diffusion Models (DMs). The core contribution is providing a density-based perspective that links the smoothness of the learned score function to a nonzero probability mass in gap regions, which mathematically leads to a positive hallucination rate.

To mitigate this, the authors introduce Variance-Guided Score Regularization (VSR), an architecture-agnostic training-time regularizer (L_smooth) that explicitly controls the score Jacobian to increase local score curvature, thereby reducing the theoretical lower bound on this "gap mass". The practical instantiation of L_smooth$ is guided by the learned denoising variance, often learned in modern DMs.

The paper also proposes two challenging new benchmark datasets, Cards and ChessImages, which feature extremely large semantic class spaces (21 x 10^5 and ~ 10^44, respectively) and employ efficient, training-free validators for systematic hallucination evaluation.

Empirical results across synthetic, standard (Hands, MNIST, Shapes), and the new challenge datasets demonstrate that VSR consistently reduces hallucinations (up to ~25% on proposed benchmarks) while maintaining or improving fidelity (FID, C-FID) and diversity/novelty (FLD).

**Strengths:**

The paper presents a **highly original and significant** contribution by establishing a novel theoretical link between **score function smoothness** and the **positive lower bound on probability mass allocated to non-data regions** (gaps), thereby formalizing the root cause of hallucinations in diffusion models.

The derived quantitative lower bound (Theorem 2) directly motivates the proposed **Variance-Guided Score Regularization (VSR)**, an **architecture-agnostic** mitigation strategy that effectively increases local score curvature via the score Jacobian. This principle-driven approach demonstrates **high quality** in its execution, evidenced by robust empirical results showing a consistent reduction in hallucination rates ($\sim 15-25\%$) across synthetic and real-world datasets, while simultaneously improving fidelity and generating novel samples (FLD). Furthermore, the introduction of the **Cards** and **ChessImages** datasets is a major **significant contribution**, providing new, scalable benchmarks with extremely high combinatorial complexity ($\sim 10^{44}$ semantic classes for ChessImages) and automated validation, addressing a key limitation in systematic hallucination evaluation for generative models. Finally, the paper's **clarity** is excellent, formalizing definitions of hallucination and clearly structuring the theoretical derivation and experimental analysis.

**Weaknesses:**

* **$\mathcal{L}_{smooth}$ Implementation Detail:** The paper mentions that calculating the high-dimensional covariance (or score Jacobian) is intractable for images and they rely on the **I-DDPM implementation to learn diagonal covariance**. While they adopt this, the **exact, explicit formulation** of the $\mathcal{L}_{smooth}$ objective *in terms of the learned diagonal covariance* for the high-dimensional image case is **not explicitly provided** in the main paper (Equation 2 uses $||J_{\theta}(x_t, t)||^2$, which is the target, but not the implementable proxy). This implementation detail is crucial for reproducibility and for confirming the practical connection between the theory and the image-based experiments.
* **Generalization vs. Memorization (ChessImages):** The paper notes that $90\%$ of valid boards are novel/generalized in the ChessImages dataset. While VSR's improvement on the FLD metric is noted, the paper should explicitly state and analyze **whether the addition of $\mathcal{L}_{smooth}$ (VSR) changes the ratio of Memorized vs. Generalized samples** compared to the baseline. Simply maintaining fidelity and reducing total hallucinations isn't enough; confirming that VSR helps promote generalization is a valuable secondary claim that needs direct evidence in the main text. Table 7 hints at this but a dedicated analysis or discussion would strengthen the claim.

**Questions:**

1.  **Please provide the explicit final formula for the Variance-Guided Score Regularization $\mathcal{L}_{smooth}$ (Equation 2) as it is implemented for the high-dimensional image datasets.** Specifically, express the term $||J_{\theta}(x_t, t)||^2$ (the squared norm of the Jacobian of the score) in terms of the **learned denoising variance/diagonal covariance** that is used in the practical implementation.
2.  **Does VSR $\mathcal{L}_{smooth}$ explicitly influence the ratio of generalized (novel) samples to memorized samples?** Using the ChessImages dataset analysis (Section 7.1), can you show how the ratio $ \frac{\text{Novel Boards}}{\text{Valid Boards}}$ changes when using the DDPM baseline versus DDPM + VSR? An explicit comparison is needed to strongly support the claim that VSR promotes the generation of novel, valid samples.
3.  The paper uses a classifier threshold of 0.98 for MNIST to detect hallucinations. **Could the authors comment on the sensitivity of the final hallucination rate (H%) to this specific threshold value?** Was a sensitivity analysis performed?

---

> ### Author Response · Authors · 2025-12-04
>
> We thanks the reviewer for their positive comments and the rating. We will address the weaknesses pointed in the revised version of manuscript.

---

### Official Review · Reviewer_25qG · 2025-11-01

**Soundness:** 2
**Presentation:** 2
**Contribution:** 3
**Rating:** 2
**Confidence:** 4

**Summary:**

This paper proposes a theoretical and practical framework for understanding and mitigating hallucinations—implausible or off-support samples—generated by diffusion models. The authors first establish a density-based analysis showing that diffusion models inevitably assign nonzero probability mass, and then introduce Variance-Guided Score Regularization (VSR), a training-time regularizer to reduce hallucinations. Empirical results across synthetic and real datasets (e.g., Hands-11K, MNIST, Shapes, and new Cards and ChessImages benchmarks) demonstrate that VSR reduces hallucination rates.

**Strengths:**

1. Elegant and practical regularization: VSR is simple to implement and architecture-agnostic.
2. Valuable benchmark contribution: The paper conducted research on a comprehensive set of datasets. Besides, Cards and ChessImages datasets expand the empirical scope for systematic hallucination evaluation at scale.

**Weaknesses:**

1. The biggest weakness of the paper is that its theorems are the unsurprising consequences of the assumptions (P1-3, A4), and the reviewer believe it is not responsible for the actual effect of the regularizer. By assuming that a smooth function has positive values on the boundary of a region, it is natural that it has non-empty integral in the interior. All the quantities (Lipschitzness, norm bounds) are merely quantities in the assumption and the paper does not show the scale of these quantities in experiments, therefore we cannot conclude that whether the assumptions are playing a significant role in inducing hallucinations in practice.

Furthermore, the theorem highlights that in order to reduce hallucination, we may need to enlarge the sharpness of the learned function in gap regions. However, the regularizer proposed in the paper encourages sharpness in the non-gap regions (training data supports). Therefore the theorem is not accountable for how the regularizer work in practice. Furthermore, in the experiments the paper is comparing models with / without regularizer with different score errors, making the comparison ineffective as score error is an important confounder for generation quality and hallucination rates.

2. The paper is badly-written in terms of mathematical rigor and result clarity. A list of confusions are listed below:
- In definition 1 the definition of H is ambiguous and probability distribution should not appear in the set definition.
- In preliminary: missing expectation symbols in ground-truth score definition; the definition support is inconsistent as it is closed but {x:p(x)>0} is open.
- In lemma 1 what is "reverse transition"? What is $\lambda^d$ on line 168? What is $P^{hall}$？
- What is hallucination rate measured in the experiments and how can it be >1 (table 2)?
- Various typos in theorems and lemmas.

3. The paper discusses the distinction between generalized samples and memorized samples, which is quite irrelevant to its main theme.

**Questions:**

Can you elaborate on the weaknesses above?

---

> ### Author Response · Authors · 2025-12-04
> **Response to reviewer-25qG.**
>
> Thanks for the reviewers for their time for insightful comments. We address the concerns below,
>
> > (W1) Unsurprising findings of the Theorem 2 and Corollary 1, and they are not responsible for the hallucinations reduction.
>
> We appreciate the concern and agree that Lemma 1 alone makes the mere existence of positive mass in the gap unsurprising; however, our use of Theorem 2 and Corollary 1 is specifically to obtain a **quantitative lower bound** on hallucination mass and to identify which geometric/analytic quantities control it. In particular, we show that the hallucination probability satisfies
>
> $$
> p_\theta(x) \ge C_b \exp\left(-S \delta_x - \frac{L}{2}\delta_x^2\right) > 0
> $$
>
> where $L$ is a local Lipschitz constant of the score (i.e., a bound on $|J_\theta|$), $S$ is a local bound on $|s_\theta|$, and $C_b = \inf_{z \in \partial G} p_\theta(z)$ is the minimum density on the boundary of the gap region. While Lemma 1 guarantees only that this quantity is strictly positive, the bound above further shows that **how large** the hallucination mass must be is directly controlled by the **curvature and magnitude of the score near the data-gap interface** and the boundary density, rather than being an arbitrary artifact of the Gaussian kernel. Intuitively, small $L$ and $S$ (an overly smooth, low-curvature score field) force a slow decay of $\log p_\theta$ across the boundary, thereby enforcing a **non-negligible** lower bound on gap mass, whereas allowing larger curvature (larger $|J_\theta|$) enables a sharper drop and thus a smaller bound. This dependence is what motivates our variance-guided score regularizer: by acting on proxies for $|J_\theta|$, it explicitly targets the quantities appearing in the bound, rather than being an unrelated heuristic. Our synthetic 1D/2D Gaussian experiments and the Hands results support this connection: we show that enforcing smoother scores (effectively reducing curvature/Lipschitzness) *increases* hallucination mass between modes (Fig. 4b), whereas our regularizer yields sharper score transitions and correspondingly **lower** hallucination rates, in line with the behavior predicted by the bound. We also see in the appendix section c that applying the regularizer increases the sharpness of the learned score and follows the ground truth score more closely (Fig 5).
>
> > (W1) Regularizer applied to all the regions.
>
> [1] show that hallucinations in diffusion models arise because the learned score provides a *smooth* approximation to a ground-truth score that has sharp “step-like’’ transitions between disjoint modes. This smoothing causes the model to interpolate between modes, assigning non-negligible probability to low-density gap regions. Conversely, when the *true* (sharp) score is used at sampling time, hallucinations essentially disappear [1].
>
> In light of this, increased score sharpness in gaps is *desirable*: it brings the learned score closer to the ideal step function and reduces the interpolation that causes hallucinations. Our regularizer is added on top of denoising score matching (DSM) as
>
> $$
> \mathcal{L} = \mathcal{L}_{DSM} + \rho \mathcal{L}_{smooth}, \rho=0.1
> $$
>
> so DSM still anchors the score to the ground-truth score function, and the regularizer only provides a mild bias among DSM-consistent solutions. We now explicitly report the range of $\rho$ in appendix section D.
>
> Figure in section C in appendix illustrates this effect. Relative to the unregularized model ($\rho=0$, dark blue), the regularized model ($\rho=0.01$, orange) (i) better matches the true score around the high-density modes (the zeros and slopes near the peaks of the empirical PDF), ensuring that in-support samples and their mean behavior are **preserved**, and simultaneously (ii) produces sharper transitions between modes that are closer to the true score in the low-density gaps. This is precisely the regime where [1] identify smooth score approximations as the source of mode interpolation. Thus, our regularizer *increases* sharpness where it matters for hallucination mitigation-boundaries-DSM ensures accurate modeling of the high-density and low-density regions. Together, this explains why we observe reduced hallucination even though the regularizer is applied globally.
>
> [1] https://arxiv.org/abs/2406.09358

---

### Meta-Review · Area_Chair_vug7 · 2025-12-19

**Summary:**

The primary reason for my decision is the unresolved disconnect between the theoretical analysis and the proposed method; Reviewers 25qG and Uy7M emphasized that while the theory motivates increasing score sharpness specifically in low-density "gap" regions to reduce hallucinations, the proposed VSR regularizer indiscriminately increases the Jacobian globally, a discrepancy the authors' rebuttal failed to justify rigorously. Additionally, Reviewer RihT noted that the foundational "disjoint support" assumption is an idealization that likely fails for high-dimensional natural images, limiting the theory's predictive power in real-world settings. These issues, combined with persistent concerns from Reviewers 25qG and Uy7M regarding mathematical ambiguities and confusing metric reporting are the leading reasons for my decision.

**Reviewer Concerns:**

The authors successfully clarified several scope and implementation issues raised by the reviewers. Specifically, they resolved Reviewer Uy7M's concern regarding the misleading introduction by explicitly stating the method targets variance-preserving models rather than flow-matching architectures like FLUX. They also addressed Reviewer fxRj's request for implementation details by linking the regularization explicitly to the diagonal covariance learned in I-DDPM.

However, critical concerns regarding the theoretical justification and experimental rigor remain unresolved. Reviewers 25qG and Uy7M highlighted a fundamental disconnect between the theory—which calls for sharpness specifically at gap boundaries—and the proposed method, which indiscriminately increases score sharpness globally; the authors' rebuttal that the primary loss "anchors" the score was deemed insufficient to rule out on-manifold distortion. Furthermore, Reviewer 25qG’s critique that the theorems are trivial consequences of the assumptions, as well as the confusion regarding metric definitions were not adequately addressed. Finally, the request by Reviewer fxRj for a specific analysis of memorization versus generalization in the ChessImages benchmark was left unfulfilled.

**Reviewer Scores:**

Reviewer 25qG: Their primary criticism regarding the disconnect between the gap-focused theory and the global regularization method was not effectively resolved by the authors' intuition-based rebuttal, and concerns about the non-triviality of the theorems remain.

Reviewer fxRj: While they strongly favored the benchmarks, the valid theoretical soundness issues raised by the other reviewers might temper their enthusiasm, though they would likely remain supportive of the paper's contributions to evaluation.

Reviewer Uy7M:  Although the authors clarified the scope regarding flow-matching models, the reviewer's fundamental objection that increasing the Jacobian globally could amplify off-manifold instability remains outstanding, as does the confusion regarding the metric reporting.

Reviewer RihT: The rebuttal confirmed that the "disjoint support" assumption is indeed an idealization, validating the reviewer's concern that the theory may lack predictive power for high-dimensional, overlapping natural image manifolds.

---

### Decision · Program_Chairs · 2026-01-26

Reject